# Improving 2D Diffusion Models for 3D Medical Imaging with Inter-Slice Consistent Stochasticity

**Chenhe Du**[1]  **Qing Wu**[1]  **Xuanyu Tian**[1]  **Jingyi Yu**[1]  **Hongjiang Wei**[2]  **Yuyao Zhang**[1][†]
[1]ShanghaiTech University  [2]Shanghai Jiao Tong University
{duchenhe, wuqing, tianxy, yujingyi, zhangyy8}@shanghaitech.edu.cn
hongjiang.wei@sjtu.edu.cn

## Abstract

3D medical imaging is in high demand and essential for clinical diagnosis and scientific research. Currently, diffusion models have become an effective tool for medical imaging reconstruction thanks to their ability to learn rich, high-quality data priors. However, learning the 3D data distribution with diffusion models in medical imaging is challenging, not only due to the difficulties in data collection but also because of the significant computational burden during model training. A common compromise is to train the diffusion model on 2D data priors and reconstruct stacked 2D slices to address 3D medical inverse problems. However, the intrinsic randomness of diffusion sampling causes severe inter-slice discontinuities of reconstructed 3D volumes. Existing methods often enforce continuity regularizations along the $z$-axis, which introduces sensitive hyper-parameters and may lead to over-smoothing results. In this work, we revisit the origin of stochasticity in diffusion sampling and introduce Inter-Slice Consistent Stochasticity (ISCS), a simple yet effective strategy that encourages inter-slice consistency during diffusion sampling. Our key idea is to control the consistency of stochastic noise components during diffusion sampling, thereby aligning their sampling trajectories without adding any new loss terms or optimization steps. Importantly, the proposed ISCS is plug-and-play and can be dropped into any 2D-trained diffusion-based 3D reconstruction pipeline without additional computational cost. Experiments on several medical imaging problems show that our method can effectively improve the performance of medical 3D imaging problems based on 2D diffusion models. Our findings suggest that controlling inter-slice stochasticity is a principled and practically attractive route toward high-fidelity 3D medical imaging with 2D diffusion priors. The code is available at: https://github.com/duchenhe/ISCS.

## 1 Introduction

Diffusion models (DMs) (Ho et al., 2020; Song et al., 2020b) have demonstrated the unparalleled ability to learn data distribution across various modalities, achieving remarkable success in both data generation and image restoration. In medical imaging, current DMs have shown effectiveness in modeling the distribution of 2D image slices. By leveraging powerful diffusion priors, DMs integrated with conventional optimization frameworks have achieved state-of-the-art (SOTA) performance in solving inverse problems in medical imaging (Song et al., 2021; Jalal et al., 2021; Chung et al., 2022; 2024; Du et al., 2024; Hu et al., 2024; Xiang et al., 2023; Wu et al., 2025; Chen et al., 2025), such as accelerated magnetic resonance imaging (MRI) and undersampled X-ray computed tomography (CT).

In clinical practice, medical data is fundamentally large 3D volumes. While individual 2D slices are crucial for review, a complete and accurate 3D volumetric reconstruction is often essential for critical downstream tasks such as precise tumor volume assessment, surgical planning, and tracking disease progression over time. However, translating the success of 2D DM-based solvers to 3D medical

---

[†]Corresponding authors.

imaging presents significant practical challenges. Due to the "***curse of dimensionality***", training DMs directly on high-dimensional volumetric data is often infeasible. The memory, computational, and data requirements for training a 3D diffusion prior are prohibitively expensive for most academic and even industrial labs (Pinaya et al., 2022; Guo et al., 2025; Wang et al., 2025). A common and practical workaround is to train a DM on 2D image slices and then apply this 2D prior in a slice-by-slice manner for 3D reconstructions (Chung et al., 2023; 2024; Chung & Ye, 2025).

While this 2D-to-3D approach mitigates the training burden, it introduces a critical new problem: *The resulting 3D volumes always lack consistency along the third dimension (i.e., z-axis).* Because each 2D slice is processed independently during the reverse diffusion process, the intrinsic randomness of the sampling procedure leads to severe inter-slice discontinuities and artifacts, degrading the quality of the final 3D volume. Existing methods have attempted to address this issue through several strategies. The most direct approach involves augmenting the 2D diffusion prior with hand-crafted regularizers, such as Total Variation (TV), to enforce smoothness between adjacent slices (Chung et al., 2024; 2023). While effective, these methods often introduce sensitive hyperparameters that require careful tuning, and can lead to over-smoothing that erases fine details. More sophisticated approaches aim to learn a more complete 3D prior, for instance by training models on 3D patches (Song et al., 2024) or by combining priors from two perpendicular 2D planes (Lee et al., 2023). Although these methods are more principled, they increase the complexity of the training or inference pipeline and may impose constraints on the data, such as requiring cubic volumes.

A similar phenomenon of discontinuity arises in the field of video restoration, where applying a pre-trained 2D image DMs to video inverse problems often leads to the lack of temporal coherence across frames. Kwon & Ye (2025; 2024) pointed out that this temporal flickering is due to the uncoordinated stochasticity inherent in the diffusion sampling process, and introduced batch-consistent sampling (BCS) to alleviate it by synchronizing stochastic noise components across frames.

In this work, we systematically investigate how the same uncoordinated stochasticity becomes the fundamental cause of inter-slice inconsistency when extending 2D diffusion priors to 3D medical reconstruction. Motivated by resulting insights, we propose **I**nter-**S**lice **C**onsistent **S**tochasticity (ISCS), a plug-and-play strategy that explicitly synchronizes the random noise component across adjacent slices using a smooth interpolation. This aligns their sampling trajectories to ensure 3D coherence without imposing excessive constraints. Extensive experiments on diverse 3D medical imaging inverse problems, including limited-view CT and MRI isotropic super-resolution (SR), demonstrate that ISCS consistently improves reconstruction quality and surpasses existing approaches, all without introducing additional loss terms, hyperparameters, or computational overhead.

## 2 PRELIMINARIES

**Diffusion Models** Diffusion models are a class of generative models that consist of a forward and a reverse process (Ho et al., 2020; Song et al., 2020b). The forward process is a fixed Markov chain that gradually adds Gaussian noise to a clean data sample $x_0 \sim p_{\text{data}}(x)$ over $T$ discrete timesteps. At each step $t$, the transition is defined as:

$$q(\mathbf{x}_t|\mathbf{x}_{t-1}) = \mathcal{N}(\mathbf{x}_t; \sqrt{1 - \beta_t}\mathbf{x}_{t-1}, \beta_t\mathbf{I}), \tag{1}$$

where $\{\beta_t\}_{t=1}^T$ is a predefined variance schedule. A key property of this process is that we can sample $\mathbf{x}_t$ directly from $\mathbf{x}_0$ in a closed form:

$$q(\mathbf{x}_t|\mathbf{x}_0) = \mathcal{N}(\mathbf{x}_t; \sqrt{\bar{\alpha}_t}\mathbf{x}_0, (1 - \bar{\alpha}_t)\mathbf{I}), \tag{2}$$

where $\alpha_t = 1 - \beta_t$ and $\bar{\alpha}_t = \prod_{i=1}^t \alpha_i$. As $t \to T$, $\mathbf{x}_T$ approaches a standard Gaussian distribution.

The reverse process learns to reverse this noising procedure, starting from a random noise sample $\mathbf{x}_T \sim \mathcal{N}(0, \mathbf{I})$ and iteratively generating a clean sample $\mathbf{x}_0$. This is achieved by training a neural network $\epsilon_\theta(\mathbf{x}_t, t)$ to predict the noise $\epsilon$ that was added to create $\mathbf{x}_t$. The network is optimized using a simplified mean squared error objective:

$$\mathcal{L}(\theta) = \mathbb{E}_{t,\mathbf{x}_0,\epsilon}\left[||\epsilon - \epsilon_\theta(\sqrt{\bar{\alpha}_t}\mathbf{x}_0 + \sqrt{1 - \bar{\alpha}_t}\epsilon, t)||_2^2\right]. \tag{3}$$

Once trained, the reverse process generates $x_{t-1}$ from $x_t$ by removing the predicted noise. Deterministic samplers like DDIM can be used to accelerate this generation process (Song et al., 2020a).

**Medical Imaging Inverse Problem** In general, the forward acquisition process of medical imaging can be formulated as follows:

$$\mathbf{y} = \mathbf{A}\mathbf{x} + \mathbf{n}, \tag{4}$$

where $\mathbf{x} \in \mathbb{R}^N$ represents the vectorized high-dimensional image to be recovered, $\mathbf{y} \in \mathbb{R}^M$ is the acquired measurement, $\mathbf{A} : \mathbb{R}^N \rightarrow \mathbb{R}^M$ is the physical-informed forward model, and $\mathbf{n} \in \mathbb{R}^M$ denotes the system noise.

It is an inverse problem to reconstruct the unknown image $\mathbf{x}$ from the observed measurement $\mathbf{y}$. For many critical applications, such as sparse-view CT or MRI isotropic super-resolution (SR), this problem is severely ill-posed (i.e., $M \ll N$), where $\mathbf{A}$ is non invertible, resulting in the analytical solution being infeasible. A robust approach is to frame the reconstruction within a Bayesian framework, seeking a Maximum A Posteriori (MAP) estimate of the image:

$$\hat{\mathbf{x}} = \arg\max_{\mathbf{x}} p(\mathbf{x}|\mathbf{y}) = \arg\min_{\mathbf{x}} \underbrace{\|\mathbf{y} - \mathbf{A}\mathbf{x}\|_2^2}_{\text{data fidelity term}} - \underbrace{\lambda \cdot \log p(\mathbf{x})}_{\text{prior term}}. \tag{5}$$

This formulation decomposes the objective into two components: a *data fidelity term*, $\|\mathbf{y} - \mathbf{A}\mathbf{x}\|_2^2$, which enforces consistency with the measurements $\mathbf{y}$, and a *prior term*, $\log p(\mathbf{x})$, which regularizes the solution by constraining it to a plausible data manifold. $\lambda$ is a weighting parameter. The power of diffusion models lies in their ability to implicitly learn a strong, complex data prior $p(\mathbf{x})$ from a training dataset. Instead of defining an explicit regularization function, they integrate this prior through a guided reverse sampling process. The update at each step is guided by the gradient of the posterior log-likelihood, which combines the unconditional score learned by the model $\nabla_{\mathbf{x}_t} \log p(\mathbf{x}_t)$ with a guidance term derived from the likelihood $\nabla_{\mathbf{x}_t} \log p(\mathbf{y}|\mathbf{x}_t)$.

**DM-based Medical Inverse Problem Solvers** The prevailing diffusion-based inverse problem solvers (DIS) integrate the data-consistency term derived from the likelihood into the reverse sampling process. To further formalize the reconstruction process, we outline a general iterative framework for solving inverse problems with a pre-trained diffusion model. The process starts with pure Gaussian noise $\mathbf{x}_T \sim \mathcal{N}(0, \mathbf{I})$ and iteratively refines the solution for each reverse timestep $t$ from $T$ down to 1. Each iteration, which aims to generate $\mathbf{x}_{t-1}$ from $\mathbf{x}_t$, can be decomposed into three fundamental steps:

*(1) Denoising Prediction*: We denote the estimate of the clean data, as $\mathbf{x}_{0|t}$, which is first predicted from the current noisy state $\mathbf{x}_t$. Leveraging Tweedie's formula, this can be achieved using the pre-trained noise prediction network $\epsilon_\theta$:

$$\mathbf{x}_{0|t} = \mathbb{E}\left[\mathbf{x}_0 \mid \mathbf{x}_t\right] = \frac{1}{\sqrt{\bar{\alpha}_t}}(\mathbf{x}_t - \sqrt{1 - \bar{\alpha}_t}\epsilon_\theta(\mathbf{x}_t)), \tag{6}$$

where $\bar{\alpha}_t$ is the noise schedule parameter at step $t$. This step effectively projects the noisy data onto the learned data manifold.

*(2) Data Fidelity Update*: The predicted clean image $\mathbf{x}_{0|t}$ is then corrected to be consistent with the physical measurements $\mathbf{y}$. This is achieved by solving a data fidelity problem, yielding an updated estimate $\hat{\mathbf{x}}_{0|t}$. This step can be expressed as a general optimization problem:

$$\hat{\mathbf{x}}_{0|t} = \arg\min_{\mathbf{z}} \|\mathbf{y} - \mathbf{A}\mathbf{z}\|_2^2 + \lambda\|\mathbf{z} - \mathbf{x}_{0|t}\|_2^2, \tag{7}$$

For many linear inverse problems, this sub-problem has a closed-form solution or can be efficiently solved using a few gradient descent steps.

*(3) Re-noising to Timestep* $t - 1$: Finally, the data-consistent estimate $\hat{\mathbf{x}}_{0|t}$ is used to generate the next iteration, $\mathbf{x}_{t-1}$. This is accomplished by applying the forward diffusion process to $\hat{\mathbf{x}}_{0|t}$, effectively injecting a controlled amount of noise corresponding to timestep $t - 1$. Taking the DDIM style sampler as an example, this process can be define as:

$$\mathbf{x}_{t-1} = \sqrt{\bar{\alpha}_{t-1}}\hat{\mathbf{x}}_{0|t} + \underbrace{\sqrt{1 - \bar{\alpha}_{t-1} - \eta^2\tilde{\beta}_t^2}\epsilon_{\theta^*}^{(t)}(\mathbf{x}_t)}_{\text{deterministic noise}} + \underbrace{\eta\tilde{\beta}_t\boldsymbol{\epsilon}}_{\text{stochastic noise}}, \tag{8}$$

where $\epsilon \sim \mathcal{N}(0, \mathbf{I})$, $\eta$ is a parameter that controls the strength of the stochastic noise component. This step re-introduces stochasticity and prepares the estimate for the subsequent iteration.

By iteratively applying these three steps, the reconstruction framework progressively refines the solution, ensuring it simultaneously adheres to the learned data prior and remains faithful to the observed measurements, thereby converging to a high-quality reconstruction.

## 3 PROPOSED METHOD

In this section, we first provide a detailed analysis of the problem's origin, identifying uncoordinated stochasticity in the slice-wise diffusion sampling process as the fundamental cause (Sec. §3.1). Then, we introduce our ISCS framework. We present its mechanism for generating correlated noise via Spherical Linear Interpolation (Slerp) and its seamless integration into existing DM-based solvers (Sec. §3.2). A geometric overview of our proposed noise strategy in comparison to conventional methods is illustrated in Fig. 1.

### 3.1 INTER-SLICE INCONSISTENCY IN 2D DM-BASED 3D RECONSTRUCTION

**Slice-wise Approximation of 3D Diffusion Prior**    The application of DMs to 3D medical imaging inverse problems presents a fundamental dilemma. Ideally, the denoising network $\epsilon_\theta$ used in the reconstruction framework (Eq. 6) should be a 3D model trained on volumetric data. However, the immense computational and data requirements for training high-quality 3D DMs make this approach often infeasible. A common alternative is to leverage a powerful, pre-trained 2D DM and apply it in a slice-by-slice manner along a chosen axis (e.g., the $z$-axis). This procedure effectively approximates the 3D denoising operation as a concatenation of independent 2D operations:

$$\tilde{\epsilon}_\theta(\mathbf{x}_t) := [\epsilon_\theta(\mathbf{x}_{t,1}), \epsilon_\theta(\mathbf{x}_{t,2}), \ldots, \epsilon_\theta(\mathbf{x}_{t,S})], \tag{9}$$

where $\mathbf{x}_{t,i}$ denotes the $i$-th slice of the noisy 3D volume $\mathbf{x}_t$.

**Root Cause of Inter-Slice Inconsistency**    While computationally efficient, this slice-wise approach introduces a significant challenge: ***inter-slice inconsistency***. The root cause of this problem is a complex interplay between the highly ill-posed nature of many medical inverse problems and the inherent stochasticity of the diffusion sampling process. Specifically, in these problems, the undersampled measurement data often fails to provide sufficient constraints to uniquely determine the solution. DMs are powerful precisely because they can fill these ambiguous regions with a strong, data-driven prior. However, this generative capability is actualized through a reverse process that, as detailed in the re-noising step (Eq. 8), repeatedly injects random noise. When this process is applied independently to each 2D slice, the combination of weak data constraints and independent stochasticity becomes highly problematic. The lack of strong guidance from the measurements gives the independently sampled noise for each slice excessive freedom to steer the sampling path. Consequently, the sampling trajectories for adjacent slices become entirely uncorrelated, leading to substantial and arbitrary variations between them. When stacked, the resulting 3D volume inevitably suffers from noticeable structural discontinuities and artifacts along the slice axis, severely degrading its diagnostic and analytical quality.

**Limitations of Post-Hoc Regularization**    A prevalent strategy to mitigate these artifacts is to apply post-hoc regularization, such as Total Variation (TV), in an additional optimization step. While effective at smoothing discontinuities, this remedy often introduces new, undesirable artifacts, such as over-smoothing or cartoon-like textures, which can erase fine diagnostic details and compromise data fidelity. More fundamentally, such methods act as an external corrective step; they do not address the intrinsic cause of the problem. They merely mask the symptoms of the underlying issue: the uncorrelated nature of the slice-wise diffusion sampling.

**Our Motivation**    This observation motivates our central research question: *Is it possible to enhance inter-slice consistency directly within the diffusion sampling process itself, obviating the need for external, artifact-inducing optimization?* We posit that the key lies in redesigning the re-noising step. A direct and intuitive strategy is to impose consistency control on the random perturbations applied across adjacent slices. Specially, instead of allowing the stochastic noise to be independent for each slice, we aim to impose a structured correlation on the random perturbations applied across the volume. By coupling the sampling paths of adjacent slices, we can fundamentally constrain the

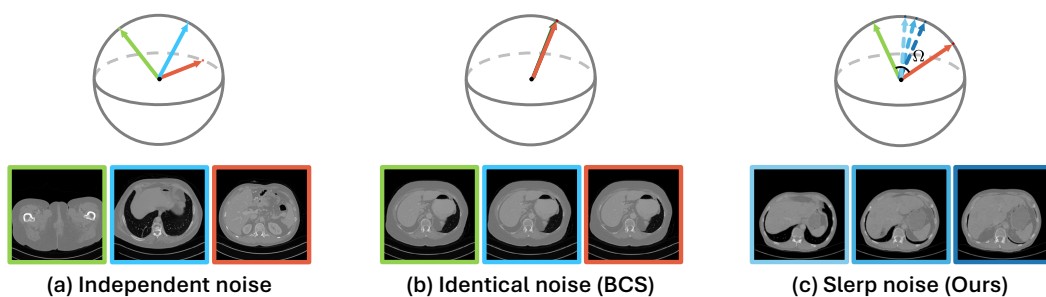

Figure 1: Geometric interpretation of how different noise strategies in the re-noising step affect the stochasticity and resulting consistency in diffusion sampling. (a) **Independent Noise** (Conventional): Independently sampled noise for each slice, leading to uncorrelated sampling paths. (b) **Identical Noise** (BCS (Kwon & Ye, 2025)): Applying the same noise to all slices forces identical sampling paths. (c) **Slerp Noise** (Ours): Our proposed ISCS interpolates noise on the hypersphere, generating smoothly correlated information across slices.

solution space and reduce stochastic inconsistencies, thereby promoting the generation of a coherent and continuous 3D volume directly from the generative process.

## 3.2 INTER-SLICE CONSISTENT STOCHASTICITY

Building on our motivation to control sampling stochasticity, we introduce a novel technique named *Inter-Slice Consistent Stochasticity (ISCS)*. The core principle of ISCS is to replace the independent Gaussian noise injection in the re-noising step (Eq. 8) with a structured, smoothly varying noise volume, which we denote as $\epsilon^{\mathrm{ISCS}}$. This modification ensures that the stochastic perturbations applied to adjacent slices are highly correlated, thereby coupling their reverse sampling trajectories. A key advantage of our approach is its simplicity; it can be seamlessly incorporated into any existing diffusion-based inverse problem solver (as shown in Sec. §2) to enforce inter-slice consistency without introducing additional optimization steps or requiring model retraining.

**Batch-Consistent Sampling & Limitations** To construct the noise volume $\epsilon^{\mathrm{ISCS}} \in \mathbb{R}^{S \times H \times W}$, we require a mechanism that generates inter-slice correlation while ensuring each slice's noise map still adheres to a standard gaussian distribution. A straightforward approach would be to apply the exact same noise to every slice. This strategy, known as batch-consistent sampling (BCS) (Kwon & Ye, 2025), has proven effective for improving temporal consistency in video inverse problems. However, naively adapting BCS to high-dimensional medical volume reconstruction is problematic. Video restoration often deals with short sequences, e.g., under 16 frames in (Kwon & Ye, 2025), where inter-frame changes are minor and dynamics are largely preserved in the measurements (Zhang et al., 2025). In contrast, medical volumes feature a much larger axial dimension (e.g., more than 300 slices for CT) with potentially significant anatomical variation between slices. Enforcing identical noise in this context is an overly rigid constraint that suppresses anatomical changes and introduces "copying artifacts," where features are improperly replicated across anatomically distinct slices.

**Correlated Noise Generation via Slerp** This motivates a more nuanced approach. We hold that an ideal noise volume should exhibit a spatially varying correlation structure. Specifically, the correlation should satisfy two key properties: (i) it should be strong between adjacent slices to ensure local consistency, and (ii) it should decay as the distance between slices increases, permitting necessary global structural divergence. In other words, we seek to define a smooth trajectory between two independent random noise instances at the boundaries of the volume. An interpolation-based strategy is therefore a natural fit. To ensuring that every interpolated state remains a valid sample from the prior $\mathcal{N}(0, \mathbf{I}_d)$, we leverage the concentration of measure phenomenon in high-dimensional spaces (Vershynin, 2018). It is a classical result that the probability mass of a high-dimensional isotropic Gaussian concentrates tightly in a thin shell around a hypersphere of radius $\sqrt{d}$. Formally, the Gaussian Annulus Theorem states that for $z \sim \mathcal{N}(0, \mathbf{I}_d)$ and any $\beta > 0$, there exists a constant $c$

such that:

$$\mathbb{P}\left(\left|\|\boldsymbol{z}\|_2 - \sqrt{d}\right| \geq \beta\right) \leq 2\exp\left(-c\beta^2\right). \tag{10}$$

This inequality implies that standard linear interpolation, which traverses the chord of the hypersphere (where $\|\boldsymbol{z}\| < \sqrt{d}$), would yield states deviating from the typical set of the distribution. Consequently, to respect the geometry of the high-probability manifold, we employ Spherical Linear Interpolation (Slerp). By tracing the geodesic path on the hypersphere surface, Slerp preserves the vector norm and distributional statistics throughout the transition.

The procedure is as follows. For a 3D volume composed of $S$ slices, we first sample two anchor noise vectors, $\mathbf{z}_1, \mathbf{z}_S \in \mathbb{R}^{H \times W}$, from the standard normal distribution $\mathcal{N}(0, \mathbf{I})$. These anchors define the start and end points of a geodesic path on the hypersphere. We then generate the noise map for each intermediate slice $\boldsymbol{\epsilon}_i^{\text{ISCS}}$ by interpolating along this path:

$$\boldsymbol{\epsilon}_i^{\text{ISCS}} = \texttt{slerp}(\mathbf{z}_1, \mathbf{z}_S; \alpha_i) = \frac{\sin((1-\alpha_i)\Omega)}{\sin(\Omega)}\mathbf{z}_1 + \frac{\sin(\alpha_i\Omega)}{\sin(\Omega)}\mathbf{z}_S, \tag{11}$$

where $\alpha_i = (i-1)/(S-1)$ is the normalized position of the $i$-th slice, and $\Omega = \arccos(\langle\mathbf{z}_1, \mathbf{z}_S\rangle/(\|\mathbf{z}_1\| \cdot \|\mathbf{z}_S\|))$ is the angle between the anchor vectors. The resulting set of noise maps $\{\boldsymbol{\epsilon}_i^{\text{ISCS}}\}_{i=1}^S$ constitutes the correlated noise volume, which varies smoothly along the slice dimension while maintaining inter-slice correlation. At the same time, each individual slice is ensured to follow a standard Gaussian distribution $\mathcal{N}(0, \mathbf{I})$.

**Integration with Diffusion Samplers**  The generated noise volume $\boldsymbol{\epsilon}^{\text{ISCS}}$ directly replaces the independently sampled noise in the re-noising step of any diffusion sampler. For instance, in the context of a DDIM-like sampler, the update rule can be formulated as:

$$\mathbf{x}_{t-1} = \sqrt{\bar{\alpha}_{t-1}}\hat{\mathbf{x}}_{0|t} + \sqrt{1 - \bar{\alpha}_{t-1} - \sigma_t^2} \cdot \boldsymbol{\epsilon}_\theta(\mathbf{x}_t) + \sigma_t \cdot \boldsymbol{\epsilon}^{\text{ISCS}}. \tag{12}$$

Here, $\hat{\mathbf{x}}_{0|t}$ is the data-consistent prediction of the clean image, the second term represents the deterministic component of the reverse step, and the final term introduces our inter-slice consistent stochasticity, with $\sigma_t$ controlling its magnitude. By construction, this process perturbs each 2D slice with similar yet smoothly distinct randomness, effectively suppressing the uncorrelated uncertainty that leads to inter-slice artifacts and thereby improving the overall consistency of the final result.

---

**Algorithm 1** 2D DIS for 3D medical imaging with ISCS

---

1: **Input:** Measurements $\mathbf{y}$, Pre-trained DM $\boldsymbol{\epsilon}_{\theta^*}$, Timesteps $T$, Noise schedule $\{\alpha_t\}_{t=0}^T$.
2: $\mathbf{x}_T \sim \mathcal{N}(0, \mathbf{I})$;
3: **for** $t = T-1, \ldots, 0$ **do**
4:     ▷ *1. Denoising Prediction*
5:     $\mathbf{x}_{0|t} = \mathbb{E}\left[\mathbf{x}_0 \mid \mathbf{x}_t\right] = \frac{1}{\sqrt{\bar{\alpha}_t}}(\mathbf{x}_t - \sqrt{1-\bar{\alpha}_t}\boldsymbol{\epsilon}_{\theta^*}(\mathbf{x}_t))$
6:     ▷ *2. Data Fidelity Update*
7:     $\hat{\mathbf{x}}_{0|t} = \arg\min_{\mathbf{z}} \|\mathbf{y} - \mathbf{A}\mathbf{z}\|_2^2 + \lambda\|\mathbf{z} - \mathbf{x}_{0|t}\|_2^2$
8:     ▷ *3. Re-noising via ISCS*
9:     $\boldsymbol{\epsilon}_i^{\text{ISCS}} = \texttt{slerp}(\mathbf{z}_1, \mathbf{z}_S; \alpha_i), \mathbf{z}_1, \mathbf{z}_S \overset{\text{i.i.d.}}{\sim} \mathcal{N}(\mathbf{0}, \mathbf{I})$
10:     $\mathbf{x}_{t-1} = \sqrt{\bar{\alpha}_{t-1}}\hat{\mathbf{x}}_{0|t} + \sqrt{1 - \bar{\alpha}_{t-1} - \eta^2\tilde{\beta}_t^2}\boldsymbol{\epsilon}_{\theta^*}^{(t)}(\mathbf{x}_t) + \eta\tilde{\beta}_t\boldsymbol{\epsilon}^{\text{ISCS}}$
11: **end for**
12: **return** $\mathbf{x}_0$

---

## 4 EXPERIMENTS

To demonstrate the effectiveness and generalization of the proposed ISCS, we conduct experiments on three typical yet challenging 3D medical imaging inverse problems: (i) **sparse-view (SV) CT reconstruction**, (ii) **limited-angle (LA) CT reconstruction**, and (iii) **MRI isotropic super-resolution (SR)**. In addition, we conduct ablation study comparing identical noise and Slerp noise to evaluate the advantages of our approach in medical 3D volume reconstruction.

Table 1: Quantitative results of compared methods and slice-to-slice difference for three 3D medical imaging tasks: SVCT of 30 views, LACT of [0, 100]°, and MRI SR of 5×. The best performance is highlighted in **bold**. $|\Delta|$ denotes the absolute gap between the inter-slice difference of the reconstruction and that of the ground truth (smaller is better; see Sec. C.1 for details).

| Task | Methods | Axial | | | Coronal | | | Sagittal | | | $|\Delta|$ |
|------|---------|-------|------|-------|---------|------|-------|----------|------|-------|------------|
| | | PSNR | SSIM | LPIPS | PSNR | SSIM | LPIPS | PSNR | SSIM | LPIPS | |
| **SVCT** | FDK | 23.91 | 0.323 | 0.324 | 23.92 | 0.414 | 0.311 | 23.79 | 0.348 | 0.310 | 0.005584 |
| | ADMM-TV | 32.94 | 0.882 | 0.113 | 33.67 | 0.895 | 0.113 | 33.72 | 0.893 | 0.107 | 0.001617 |
| | DDNM | 32.55 | 0.851 | 0.076 | 32.51 | 0.837 | 0.227 | 32.79 | 0.834 | 0.200 | 0.009342 |
| | DDNM+ISCS | 33.97 | 0.885 | 0.076 | 33.53 | 0.896 | 0.114 | 33.97 | 0.893 | 0.106 | 0.001785 |
| | DDS | 34.76 | 0.919 | 0.069 | 35.12 | 0.906 | 0.149 | 35.33 | 0.904 | 0.141 | 0.005588 |
| | DDS+TV | 36.26 | 0.931 | 0.073 | 37.08 | 0.938 | 0.095 | 37.50 | 0.936 | 0.088 | 0.002937 |
| | DDS+ISCS | **36.97** | **0.937** | **0.064** | **37.75** | **0.944** | **0.070** | **38.16** | **0.942** | **0.065** | 0.001835 |
| **LACT** | FDK | 15.85 | 0.381 | 0.233 | 16.30 | 0.453 | 0.240 | 16.71 | 0.413 | 0.223 | 0.000898 |
| | ADMM-TV | 27.27 | 0.795 | 0.125 | 27.85 | 0.794 | 0.123 | 27.83 | 0.799 | 0.120 | 0.000877 |
| | DDNM | 28.40 | 0.854 | 0.076 | 28.75 | 0.774 | 0.245 | 28.22 | 0.775 | 0.235 | 0.016443 |
| | DDNM+ISCS | 30.89 | 0.898 | **0.066** | 31.88 | 0.906 | 0.084 | 31.59 | 0.908 | 0.076 | 0.001899 |
| | DDS | 29.07 | 0.885 | 0.086 | 29.93 | 0.829 | 0.196 | 29.20 | 0.830 | 0.193 | 0.011592 |
| | DDS+TV | 31.40 | 0.898 | 0.086 | **33.33** | 0.906 | 0.110 | **32.83** | 0.909 | 0.104 | 0.002566 |
| | DDS+ISCS | **31.65** | **0.911** | 0.071 | 32.90 | **0.917** | **0.082** | 32.49 | **0.920** | **0.077** | 0.001966 |
| **MRI SR** | Cubic | 38.02 | 0.925 | 0.032 | 36.22 | 0.908 | 0.095 | 36.85 | 0.904 | 0.111 | 0.004168 |
| | ADMM-TV | 39.34 | 0.940 | 0.044 | 38.27 | 0.934 | 0.060 | 36.86 | 0.906 | 0.097 | 0.005074 |
| | DDNM | 38.32 | 0.954 | 0.051 | 38.83 | 0.954 | 0.039 | 37.84 | 0.933 | 0.068 | 0.001480 |
| | DDNM+ISCS | 39.62 | 0.963 | **0.019** | 39.84 | 0.959 | **0.022** | 38.65 | 0.939 | **0.045** | 0.001913 |
| | DDS | 39.32 | 0.952 | 0.038 | 38.84 | 0.951 | 0.027 | 37.94 | 0.925 | 0.082 | 0.001853 |
| | DDS+TV | 40.12 | 0.958 | 0.031 | 39.36 | 0.955 | 0.038 | 38.53 | 0.932 | 0.075 | 0.004732 |
| | DDS+ISCS | **40.33** | **0.968** | **0.019** | 39.84 | **0.965** | 0.035 | **39.35** | **0.948** | 0.052 | 0.002096 |

## 4.1 EXPERIMENTAL SETTINGS

**CT Dataset & Pre-processing** We use the AAPM 2016 low-dose CT grand challenge dataset (Mc-Collough et al., 2017), which contains 5936 CT slices from 10 patients. All slices are first resized to 256×256. We use 5410 slices from 9 patients for training the diffusion model, and reserve one patient's data (L506) for evaluation. The size of the evaluation volume is 256×256×300. We employ torch-radon[1] library with cone-beam (CB) geometry to simulate projections. For sparse-view, we sample 30 views uniformly from $[0°, 360°)$; for limited-angle, 100 views from $[0°, 100°]$. *Detailed CBCT geometry setting can be found in the Appendix.*

**MRI Dataset & Pre-processing** We use the public IXI dataset[2] which contains multiple modality human brain scans. For our experiments, we use the T1-weighted images. The evaluation volume has a size of 256×256×150 with voxel spacing of $1.2×0.9375×0.9375$ mm$^3$. We first resample the data to isotropic 1 mm$^3$ resolution, then pad it to a size of 256×256×256. To simulate anisotropic scans, we apply a 5× downsampling along the $z$-axis.

**Implementation Details** For the diffusion prior, we adopt the Variance Exploding (VE) diffusion model Song et al. (2020b), following the architecture in Chung et al. (2023). For the CT reconstruction task, the model is trained on the AAPM dataset as described above, while for the MRI isotropic SR task, we use the pretrained checkpoint trained on coronal axis slice provided by Lee et al. (2023). To ensure a fair comparison across different DIS methods, we employ 30 NFEs for CT and 20 NFEs for MRI throughout all experiments, all DIS methods use the same pre-trained diffsuion prior.

**Methods in Comparison & Metrics** For 3D medical inverse problems, we compared two representative state-of-the-art (SOTA) DIS methods: DDNM (Wang et al., 2022) and DDS (Chung et al., 2024). Since our approach is DM-agnostic and can be seamlessly integrated into existing frameworks,

---

[1] https://github.com/carterbox/torch-radon
[2] https://brain-development.org/ixi-dataset

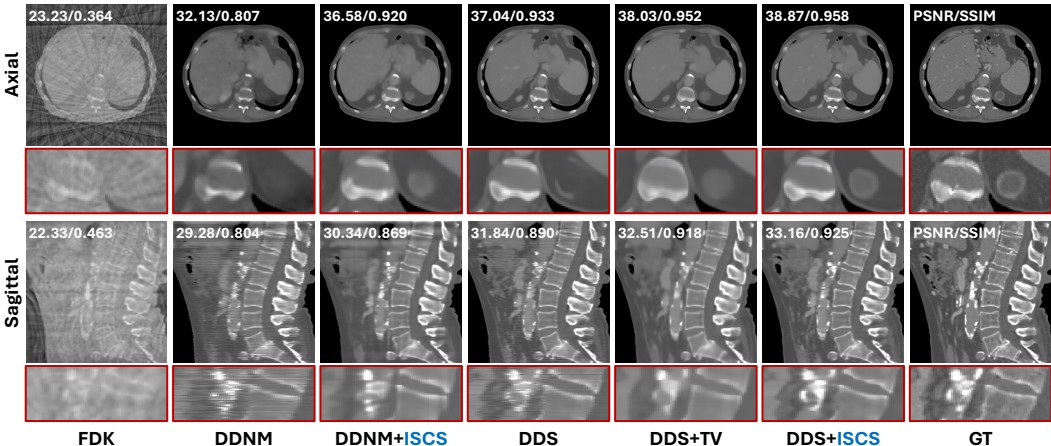

Figure 2: Qualitative results of compared methods on a representative sample for SVCT of 30 views. The display window is set as [-480, 820] HU.

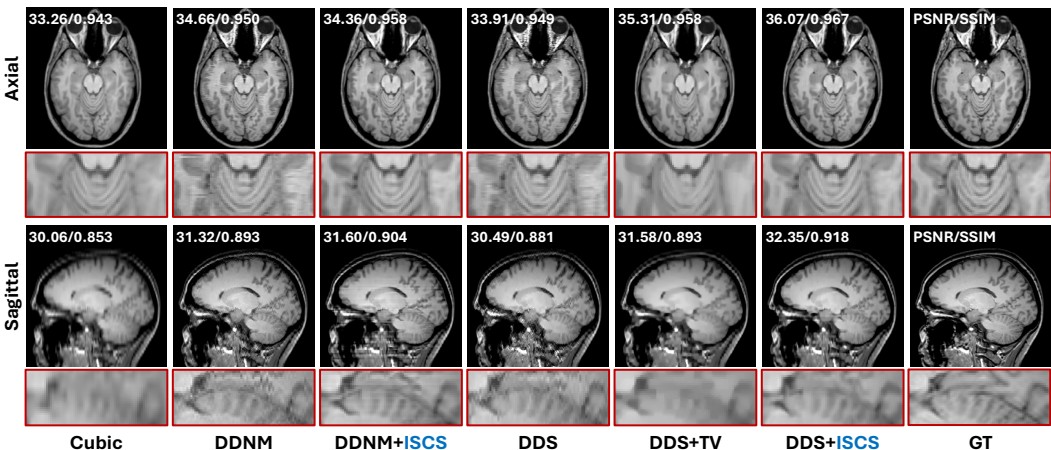

Figure 3: Qualitative results of compared methods on a representative sample for MRI SR of $5\times$.

we denote the variants as DDNM+ISCS and DDS+ISCS. In addition, we compare with traditional reconstruction baselines: FDK and ADMM-TV for CT reconstruction, and cubic upsampling and ADMM-TV for MRI super-resolution. For quantitative evaluation of reconstructed volumes, we adopt two classical data-fidelity metrics, peak signal-to-noise ratio (PSNR) and structural similarity index measure (SSIM), and one perceptual metric, LPIPS (Zhang et al., 2018).

## 4.2 MAIN RESULTS

**SVCT and LACT** Table 1 shows the quantitative results of the compared methods with and without the proposed ISCS strategy. Incorporating ISCS into both DDNM and DDS yields consistent improvements across all three views, in some cases even surpassing TV regularization that relies on additional optimization steps. The gains are particularly pronounced in SSIM and LPIPS for the coronal and sagittal planes, indicating that ISCS effectively mitigates inter-slice discontinuities along the z-axis while preserving structural fidelity. Representative qualitative results on SVCT and LACT are shown in Fig. 2 and Fig. 8. Without any constraint on inter-slice continuity, DDNM and DDS produce acceptable reconstructions on axial slices but suffer from severe discontinuities and fragmented artifacts in sagittal views due to independent stochasticity. Although adding TV regularization alleviates these artifacts, it introduces blurring and cartoon-like artifacts that degrade fine anatomical details. In contrast, ISCS markedly reduces inter-slice inconsistency in both DDNM

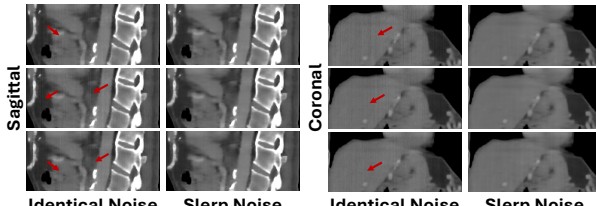

Figure 4: Qualitative results of adopting identical (BCS) and slerp noise (ISCS) during re-noising, where the red arrows denote the noticeable "copying artifacts". The display window is set as [-480, 820] HU.

Table 2: Quantitative results of adopting identical (BCS) and slerp noise (ISCS) during re-noising.

| Noise Type | | BCS | ICSC |
|---|---|---|---|
| **Coronal** | PSNR | 38.00 | **38.16** |
| | SSIM | 0.937 | **0.941** |
| | LPIPS | 0.081 | **0.074** |
| **Sagittal** | PSNR | 38.24 | **38.78** |
| | SSIM | 0.933 | **0.937** |
| | LPIPS | 0.081 | **0.073** |

and DDS without introducing such artifacts, preserving sharp edges and yielding reconstructions that are most consistent with the GT.

**MRI Isotropic SR**   Quantitative comparisons and qualitative visualizations are presented in Table 1 and Fig. 3, respectively. The compared methods exhibit trends consistent with those observed in the CT task when evaluated with and without the proposed ISCS strategy. It is worth noting that the pretrained DM we adopt was trained on coronal slices, so inter-slice inconsistencies mainly manifest in the axial and sagittal views. The results show that ISCS substantially alleviates these inconsistencies without introducing blurring artifacts. In particular, the zoomed sagittal view of the cerebellum in Fig. 3 highlights that, without any inter-slice constraint, both DDNM and DDS produce noticeable streak-like artifacts, whereas TV-based regularization suppresses these artifacts at the cost of oversmoothing structural details. By contrast, incorporating the proposed ISCS yields reconstructions that best preserve fine anatomical structures and are most consistent with the GT.

### 4.3 DISCUSSIONS

**Effectiveness of Correlated Noise Generation via Slerp**   We examine the impact of the proposed ISCS strategy (Slerp-based noise) and the BCS strategy (Kwon & Ye, 2025) (identical noise) on reconstruction performance in Eq. 12. Under the same experimental setting, each strategy is run five times. Quantitative results are reported in Table 2 and visualizations in Fig. 4. The results of BCS exhibit noticeable streak artifacts along the z-axis. This observation is consistent with the analysis in Sec. § 3.2, suggesting that in high-dimensional medical volumes, enforcing identical stochasticity across all slices suppresses natural structural variations and leads to replicated artifacts.

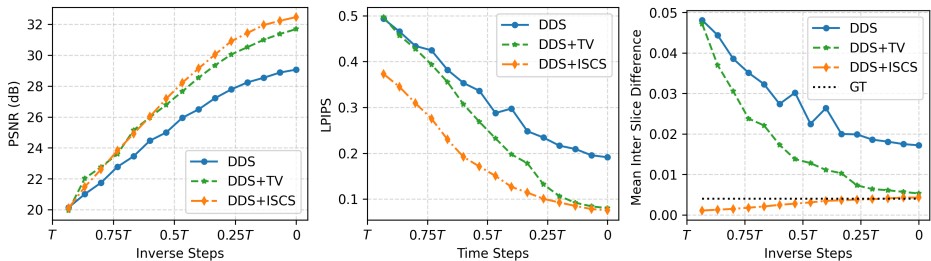

Figure 5: Performance curves across the sampling process, where a higher PSNR and lower LPIPS and inter-slice difference reflect improved data fidelity and better inter-slice consistency.

**Trajectory of the sampling process**   Fig. 5 plots the evolution of performance over diffusion time steps $T$ (i.e., intermediate reconstructions as $T \to 0$). With ISCS, the inter-slice difference drops close to the ground-truth reference early in the trajectory, while PSNR/LPIPS continue to improve thereafter. Compared with the DDS baseline, which maintains larger inter-slice gaps until later steps, ISCS achieves cross-slice coherence earlier and sustains it across sampling. This early stabilization may reduce the effective search space and help the sampler converge more reliably.

## 5 CONCLUSION

We identify uncoordinated stochasticity in diffusion sampling as a key driver of inter-slice inconsistency when 2D diffusion priors are applied to 3D medical reconstruction. To address this, we introduce ISCS, a plug-and-play strategy that correlates per-slice noise via spherical linear interpolation, aligning sampling trajectories without adding loss terms, hyperparameters, retraining, or extra compute. In our experiments on SVCT, LACT and MRI isotropic SR, ISCS generally yields better quantitative scores and smoother cross-view continuity while largely preserving edges and fine anatomy. We view stochasticity control as a practical path to narrow the gap between 2D priors and 3D fidelity, and we plan to explore learned, data-adaptive correlation fields and integration with multi-plane or 3D priors.

### ACKNOWLEGEMENT

This work was supported by the National Natural Science Foundation of China (Grant No. 62571328) and, in part, by the National Natural Science Foundation of China (Grant No. W2431046), the MoE Key Lab of Intelligent Perception and Human–Machine Collaboration (ShanghaiTech University), and the Shanghai Frontiers Science Center of Human-centered Artificial Intelligence.

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

## A  APPENDIX

### A.1  USE OF LLMs

We affirm that our work was conducted in accordance with the ICLR guidelines regarding the use of large language models (LLMs). LLMs were utilized solely for minor text editing, specifically for correcting spelling and grammar errors. No LLM was used to generate or alter any core technical content, including the research methodology, experimental design, results, or conclusions. The ideas, data analysis, and scientific conclusions presented in this paper are the sole product of the authors' original work.

### A.2  ADDITIONAL DETAILS OF EXPERIMENT

**CT Datasets & Pre-processing**  To standardize the input data, the Hounsfield Unit (HU) values were clipped to the range $[-1000, 1600]$ and linearly rescaled to $[0, 1]$. For the cone-beam geometry, we adopt a configuration with detector dimensions of $512 \times 512$, detector spacing of 2 mm in both $u$- and $v$-directions, source-to-isocenter distance of 700 mm, and source-to-detector distance of 500 mm.

#### A.2.1  NETWORK TRAINING

A VE-form (Song et al., 2020b) diffusion model based on the `ncsnpp` architecture was trained on the AAPM dataset for 70 epochs with a batch size of 16 and a learning rate of $2 \times 10^{-4}$, using a single NVIDIA A100 GPU. The model was optimized with Adam ($\beta_1 = 0.9$, $\epsilon = 10^{-8}$) without weight decay.

### A.3  ADDITIONAL DETAILS OF BASELINES

**ADMM-TV**  Following the protocol of Chung et al. (2023), the solution is to optimize the following objective:

$$\mathbf{x}^* = \arg\min_{\mathbf{x}} \frac{1}{2} \|\mathbf{A}\mathbf{x} - \mathbf{y}\|_2^2 + \lambda \|\boldsymbol{D}\mathbf{x}\|_{2,1}, \tag{13}$$

where $\boldsymbol{D} := [\mathbf{D}_x, \mathbf{D}_y, \mathbf{D}_z]$, and is solved with standard ADMM and CG. Hyper-parameters are set identical to Chung et al. (2023).

**DDNM (Wang et al., 2022)**  DDNM enforces measurement consistency through a range–null space decomposition. The update rule can be expressed as

$$\mathbb{E}[\mathbf{x}_0 \mid \mathbf{x}_t, \mathbf{y}]^{(\mathrm{DDNM})} \approx \left(\mathbf{I} - \mathbf{A}^\dagger \mathbf{A}\right) D_{\theta^*}(\mathbf{x}_t) + \mathbf{A}^\dagger \mathbf{y}, \tag{14}$$

where $\mathbf{A}^\dagger$ denotes the pseudo-inverse of the measurement operator $\mathbf{A}$. For CT reconstruction, SIRT is adopted as the pseudo-inverse operator for stability.

**DDS (Chung et al., 2024)**  DDS can be interpreted as solving the following proximal optimization problem:

$$\mathbb{E}[\mathbf{x}_0 \mid \mathbf{x}_t, \mathbf{y}]^{(\mathrm{DDS})} \approx \arg\min_{\mathbf{x}_0} \frac{\gamma}{2} \|\mathbf{y} - \mathbf{A}\mathbf{x}_0\|_2^2 + \frac{1}{2} \|\mathbf{x}_0 - D_{\theta^*}(\mathbf{x}_t)\|_2^2, \tag{15}$$

which is equivalent to solving the following linear system:

$$\left(\gamma \mathbf{A}^\top \mathbf{A} + \mathbf{I}\right) x_0^* = \gamma \mathbf{y} + D_{\theta^*}(\mathbf{x}_t), \tag{16}$$

typically using the conjugate gradient (CG) method. In our experiments, the number of CG iterations was set to 10, which we found to be the most suitable value based on tuning with the validation data.

## B    EXTENDED THEORETICAL ANALYSIS

In this section, we deepen the theoretical grounding of the proposed Inter-Slice Consistency Sampling (ISCS). Specifically, we analyze the geometric stability of our stochastic interpolation strategy and discuss the compatibility of ISCS with deterministic sampling trajectories (ODEs).

### B.1    GEOMETRIC STABILITY OF RANDOM ANCHOR SELECTION

A core component of ISCS is the generation of spatially correlated noise via interpolation between anchor noise vectors. A natural question arises regarding the sensitivity of the method to the random selection of these anchors. Here, we provide a theoretical justification for using independent random sampling as a robust default.

**Proposition B.1** (Concentration of Angular Distance in High Dimensions). *Let $z_1, z_2 \sim \mathcal{N}(0, I_d)$ be two independent random vectors in $\mathbb{R}^d$. As the dimension $d \to \infty$, the angle $\theta(z_1, z_2)$ between them converges in probability to $\pi/2$:*

$$\lim_{d \to \infty} P\left( \left| \theta(z_1, z_2) - \frac{\pi}{2} \right| \geq \epsilon \right) = 0, \quad \forall \epsilon > 0. \tag{17}$$

*Remark* B.1. **(Implication for ISCS)** Since CT reconstruction typically involves high-dimensional latent spaces ($d \gg 1$), any two independently sampled anchor vectors are approximately orthogonal ($\approx 90°$) with high probability. This concentration of measure phenomenon implies that the "random" selection of anchors actually yields a geometrically consistent interpolation path length across different runs. Consequently, our method is inherently stable and does not require hyperparameter tuning for the angular separation of anchors, a claim we validate empirically in Section C.4.

### B.2    COMPATIBILITY WITH DETERMINISTIC SAMPLERS

While diffusion models often employ stochastic samplers, ISCS is fundamentally compatible with deterministic ODE solvers (e.g., DDIM with $\eta = 0$).

Let the reverse diffusion process be modeled as an ODE $d\mathbf{x}_t = f(\mathbf{x}_t, t)dt$. The trajectory of this ODE is uniquely determined by the terminal condition $\mathbf{x}_T$. Although deterministic sampling removes stochasticity from the reverse denoising trajectory, the entire reconstruction process remains fully conditioned on the choice of initial latent $\mathbf{x}_T$. By applying ISCS to correlate the initial noise volume $\mathbf{x}_T$ across slices, we enforce that the starting points of the reverse process lie on a spatially smooth manifold. Since the subsequent integration steps are deterministic maps, this initial coherence is preserved throughout the trajectory. We provide empirical verification of this property in Section C.3.

## C    ADDITIONAL EMPIRICAL RESULTS

We present comprehensive experiments to evaluate the robustness of ISCS against 3D-aware baselines, its performance under deterministic sampling, and its sensitivity to anchor selection.

### C.1    QUANTITATIVE EVALUATION OF INTER-SLICE CONSISTENCY

To provide a holistic evaluation beyond standard 2D metrics (PSNR, SSIM), we introduce a dedicated metric to quantify inter-slice consistency while accounting for natural anatomical variations.

**Metric Definition**    We define the Slice Difference (SDiff) as the mean absolute difference between adjacent slices along the z-axis:

$$\text{SDiff} = \frac{1}{S-1} \sum_{i=1}^{S-1} \text{Mean}(|x_{i+1} - x_i|), \tag{18}$$

where $S$ is the total number of slices and $x_i$ represents the $i$-th slice.

Crucially, simply minimizing SDiff is insufficient, as it favors over-smoothed solutions (where SDiff $\to 0$) that lack texture. Therefore, to evaluate fidelity, we report the **Absolute Gap** ($|\Delta|$)

between the reconstruction and the Ground Truth (GT):

$$|\Delta| = |\text{SDiff}_{\text{recon}} - \text{SDiff}_{\text{GT}}|. \tag{19}$$

A smaller $|\Delta|$ indicates that the reconstructed volume successfully replicates the natural anatomical coherence of the GT, rather than enforcing artificial smoothness.

**Results Analysis** As detailed in Table 1 and Table 3, our method consistently achieves the lowest $|\Delta|$ compared to baselines. This quantitative evidence confirms that ISCS not only mitigates flickering artifacts but also restores 3D coherence to a level that closely matches real anatomy, avoiding the over-smoothing pitfalls common in explicit regularization methods.

Table 3: Slice-to-slice difference of compared methods for three 3D medical imaging tasks: SVCT of 30 views, LACT of $[0, 100]°$, and MRI SR of $5\times$. $\text{SDiff}_{\text{recon}}$ and $\text{SDiff}_{\text{GT}}$ denote the mean absolute difference between adjacent slices for the reconstruction and ground truth, respectively; $\Delta = \text{SDiff}_{\text{recon}} - \text{SDiff}_{\text{GT}}$ measures the signed gap, and $|\Delta|$ is its absolute value (smaller is better).

| Task | Methods | Slice difference | | | |
|------|---------|------------------|-----|---|---|
| | | $\text{SDiff}_{\text{recon}}$ | $\text{SDiff}_{\text{GT}}$ | $\Delta$ | $|\Delta|$ |
| **SVCT** | FDK | 0.011893 | 0.006309 | 0.005584 | 0.005584 |
| | ADMM-TV | 0.004692 | 0.006309 | -0.001617 | 0.001617 |
| | DDNM | 0.015652 | 0.006309 | 0.009342 | 0.009342 |
| | DDNM+ISCS | 0.004524 | 0.006309 | -0.001785 | 0.001785 |
| | DDS | 0.011897 | 0.006309 | 0.005588 | 0.005588 |
| | DDS+TV | 0.003372 | 0.006309 | -0.002937 | 0.002937 |
| | DDS+ISCS | 0.004474 | 0.006309 | -0.001835 | 0.001835 |
| **LACT** | FDK | 0.007208 | 0.006309 | 0.000898 | 0.000898 |
| | ADMM-TV | 0.005432 | 0.006309 | -0.000877 | 0.000877 |
| | DDNM | 0.022753 | 0.006309 | 0.016443 | 0.016443 |
| | DDNM+ISCS | 0.004410 | 0.006309 | -0.001899 | 0.001899 |
| | DDS | 0.017901 | 0.006309 | 0.011592 | 0.011592 |
| | DDS+TV | 0.003743 | 0.006309 | -0.002566 | 0.002566 |
| | DDS+ISCS | 0.004344 | 0.006309 | -0.001966 | 0.001966 |
| **MRI SR** | Cubic | 0.007766 | 0.011934 | -0.004168 | 0.004168 |
| | ADMM-TV | 0.006861 | 0.011934 | -0.005074 | 0.005074 |
| | DDNM | 0.013414 | 0.011934 | 0.001480 | 0.001480 |
| | DDNM+ISCS | 0.010021 | 0.011934 | -0.001913 | 0.001913 |
| | DDS | 0.013787 | 0.011934 | 0.001853 | 0.001853 |
| | DDS+TV | 0.007202 | 0.011934 | -0.004732 | 0.004732 |
| | DDS+ISCS | 0.009838 | 0.011934 | -0.002096 | 0.002096 |

## C.2 COMPARISON WITH EXPLICIT 3D-AWARE PRIORS

We compare our method against state-of-the-art 3D-aware diffusion priors, specifically Diffusion-Blend (Song et al., 2024) and TPDM (Lee et al., 2023). While 3D-aware methods generally offer strong performance in ideal settings, our results highlight distinct trade-offs regarding flexibility, robustness, and computational complexity.

**Experimental Setup** To ensure a fair comparison, all priors were trained on the same AAPM CT dataset (1mm slice thickness). We evaluate on two distinct subjects to test generalization:

- **Subject L506 (In-Distribution):** Standard 1mm thickness ($300 \times 256 \times 256$), matching the training distribution.
- **Subject L221 (Out-of-Distribution):** Thick 5mm slices ($99 \times 256 \times 256$), representing a significant domain shift.

Note that TPDM requires cubic volumes ($D \geq 256$) for orthogonal consistency checks and thus cannot be applied to Subject L221 ($D = 99$).

Table 4: Quantitative comparison on the SVCT task of 30 views. Higher PSNR/SSIM and lower LPIPS indicate better performance.

| Subject | Method | Axial | | | Coronal | | | Sagittal | | |
|---|---|---|---|---|---|---|---|---|---|---|
| | | PSNR | SSIM | LPIPS | PSNR | SSIM | LPIPS | PSNR | SSIM | LPIPS |
| **L506** (1mm) | DDS | 34.76 | 0.919 | 0.069 | 35.12 | 0.906 | 0.149 | 35.33 | 0.904 | 0.141 |
| | DDS+ISCS | 36.97 | 0.937 | 0.064 | 37.75 | 0.944 | 0.070 | 38.16 | 0.942 | 0.065 |
| | TPDM | 37.59 | **0.944** | 0.063 | 38.40 | **0.950** | 0.068 | 38.64 | **0.948** | 0.062 |
| | DiffusionBlend | **38.22** | 0.943 | **0.034** | **38.95** | 0.945 | **0.047** | **39.29** | 0.943 | **0.042** |
| **L221** (5mm) | DDS | 37.83 | 0.952 | 0.045 | 37.96 | 0.952 | 0.055 | 38.62 | 0.950 | 0.045 |
| | DDS+ISCS | 38.70 | **0.956** | **0.045** | 39.03 | **0.963** | **0.037** | 39.70 | **0.961** | **0.031** |
| | TPDM | – | – | – | – | – | – | – | – | – |
| | DiffusionBlend | **38.84** | 0.944 | 0.046 | **39.05** | 0.946 | 0.054 | 39.44 | 0.946 | 0.045 |

Table 5: Quantitative comparison on the LACT task of [0, 100]°. Higher PSNR/SSIM and lower LPIPS indicate better performance.

| Subject | Method | Axial | | | Coronal | | | Sagittal | | |
|---|---|---|---|---|---|---|---|---|---|---|
| | | PSNR | SSIM | LPIPS | PSNR | SSIM | LPIPS | PSNR | SSIM | LPIPS |
| **L506** (1mm) | DDS | 29.03 | 0.885 | 0.086 | 29.84 | 0.828 | 0.197 | 29.13 | 0.829 | 0.194 |
| | DDS+ISCS | **31.65** | 0.911 | 0.071 | 32.90 | 0.917 | 0.082 | 32.49 | 0.920 | 0.077 |
| | TPDM | 30.95 | 0.912 | 0.062 | 32.48 | **0.920** | 0.070 | 32.49 | **0.924** | 0.064 |
| | DiffusionBlend | 31.11 | **0.915** | **0.037** | **33.16** | 0.917 | **0.053** | **32.76** | 0.922 | **0.049** |
| **L221** (5mm) | DDS | 29.15 | 0.895 | 0.053 | 29.94 | 0.860 | 0.139 | 30.12 | 0.862 | 0.124 |
| | DDS+ISCS | 30.45 | 0.908 | 0.042 | 31.42 | 0.910 | 0.053 | 31.85 | 0.912 | 0.046 |
| | TPDM | – | – | – | – | – | – | – | – | – |
| | DiffusionBlend | **31.10** | **0.918** | **0.029** | **32.94** | **0.921** | **0.045** | **33.02** | **0.923** | **0.037** |

**Results Analysis**  Quantitative comparisons for SVCT and LACT tasks are presented in Table 4 and Table 5, respectively.

1. **Performance on Ideal Data:** On Subject L506, explicit 3D methods (DiffusionBlend, TPDM) generally outperform the 2D baseline (DDS), confirming the value of 3D priors. However, ISCS significantly narrows this gap, recovering high-frequency details without requiring 3D architectural changes.

2. **Robustness to Domain Shift:** On Subject L221 (5mm slices), the advantage of 3D-aware methods diminishes due to the mismatch between the training prior (1mm) and test data (5mm). Notably, ISCS achieves comparable or superior performance (e.g., higher SSIM in SVCT), demonstrating that regulating stochasticity offers greater flexibility than enforcing rigid 3D spatial constraints.

While 3D-aware methods offer high performance, they necessitate task-specific network designs, 3D-aware architectures, and additional training, which significantly increases computational cost and engineering complexity. ISCS focuses on a different axis of improvement: enhancing 2D priors through sampling-time noise correlation without architectural changes. Crucially, ISCS is orthogonal to 3D-aware methods. In principle, ISCS can be integrated into frameworks like DiffusionBlend or TPDM to further enhance their consistency, a direction we plan to explore in future work.

## C.3 ANALYSIS OF ISCS WITH DETERMINISTIC SAMPLERS

We investigate the interaction between the proposed ISCS and the stochasticity of the sampler (i.e., DDIM parameter $\eta$).

**Effectiveness on Deterministic Samplers** ($\eta = 0$)  As discussed in Section B.2, ISCS is applicable to deterministic paths by correlating the initialization. We conducted an evaluation on the SVCT task with 30 views using DDIM with $\eta = 0$. As summarized in Table 6, incorporating ISCS into the initialization phase leads to substantial performance gains over independent per-slice initialization.

Improvements are particularly pronounced in the Coronal and Sagittal orientations: for instance, Coronal LPIPS decreases from 0.239 to 0.065, and PSNR increases from 32.67 to 37.10. These results demonstrate that ISCS alleviates slice-wise discontinuities even when the sampling dynamics themselves are deterministic.

**Necessity of Stochasticity** Despite the compatibility with $\eta = 0$, further analysis indicates that deterministic sampling is not optimal for the inverse problems considered in this work. Table 7 reports an ablation over $\eta$ values in DDIM. Reconstruction quality improves monotonically as $\eta$ increases, with the fully stochastic setting ($\eta \approx 1$) achieving the best PSNR and SSIM. This observation is consistent with recent findings in diffusion-based inverse problems (Kwon & Ye, 2025; Zhu et al., 2023; Nie et al., 2024; Wang et al., 2022; Kawar et al., 2022), which emphasize the importance of controlled stochasticity for escaping local minima and recovering high-frequency structures. Accordingly, although ISCS remains effective at $\eta = 0$, the main experiments in the manuscript adopt stochastic sampling to obtain the best overall reconstruction fidelity.

Table 6: **ISCS with Deterministic Sampling.** Effect of ISCS on initial noise under DDIM $\eta = 0$ (SVCT-30). Significant improvements in auxiliary views demonstrate successful consistency enforcement.

|  | PSNR | SSIM | LPIPS |
|---|---|---|---|
| **Axial** | | | |
| DDIM ($\eta = 0$) | 32.30 | 0.864 | 0.065 |
| + ISCS | **36.19** | **0.924** | **0.053** |
| **Coronal** | | | |
| DDIM ($\eta = 0$) | 32.67 | 0.806 | 0.239 |
| + ISCS | **37.10** | **0.935** | **0.065** |
| **Sagittal** | | | |
| DDIM ($\eta = 0$) | 32.93 | 0.805 | 0.228 |
| + ISCS | **37.58** | **0.932** | **0.059** |

Table 7: **Stochasticity Ablation.** Performance vs. $\eta$ on SVCT-30. Higher stochasticity yields better reconstruction.

| $\eta$ | PSNR | SSIM |
|---|---|---|
| 0.0 | 34.48 | 0.916 |
| 0.1 | 34.56 | 0.918 |
| 0.2 | 34.77 | 0.923 |
| 0.3 | 35.06 | 0.927 |
| 0.4 | 35.38 | 0.932 |
| 0.5 | 35.70 | 0.936 |
| 0.6 | 35.99 | 0.939 |
| 0.7 | 36.26 | 0.942 |
| 0.8 | 36.52 | 0.945 |
| 0.9 | 36.77 | 0.948 |
| **1.0** | **37.08** | **0.951** |

## C.4 STABILITY OF ANCHOR SELECTION IN ISCS

This section analyzes the stability of ISCS with respect to the choice of anchor noise vectors $(z_1, z_S)$ and random seeds. As shown in Proposition B.1, the angle between two independently sampled Gaussian vectors concentrates sharply around $90°$ with vanishing variance. Consequently, independent random sampling of anchor vectors naturally yields a consistent path length across runs, implying that the ISCS interpolation manifold is robust to variations in $z_1$ and $z_S$.

To further quantify this robustness, we perform an ablation in which the angle between $z_1$ and $z_S$ is no longer random but fixed to prescribed values $\theta \in \{30°, 60°, 90°, 120°, 150°, 175°\}$. For each angular constraint, we generate 10 independent trials on the SVCT-30 task. The results, summarized in Table 8, demonstrate highly consistent performance across the entire angular spectrum. The fluctuations in PSNR, SSIM, and LPIPS remain extremely small, with standard deviations mostly below $0.2\,\text{dB}$. Notably, even the difference between the lowest- and highest-performing settings (e.g., $150°$ vs. $175°$) remains under $0.15\,\text{dB}$ in the Axial view. These results confirm that ISCS exhibits strong invariance to the anchor angle and random seeds, validating the use of simple independent sampling (yielding $\theta \approx 90°$ by default) as a robust and parameter-free design choice.

## C.5 SCALABILITY AND GENERALIZATION ASSESSMENT

In this section, we critically evaluate the scalability and generalization capabilities of ISCS, focusing on two challenging scenarios: pathological cases with abrupt structural changes and data acquisition with varying slice thicknesses.

Table 8: **Stability of Anchor Selection Strategy.** Quantitative results (Mean $\pm$ Std) of ISCS under different anchor angles (10 independent runs each) on SVCT-30 task. The method shows minimal sensitivity to the specific geometric configuration of the latent noise.

| | Axial | | | Coronal | | | Sagittal | | |
|---|---|---|---|---|---|---|---|---|---|
| Angle ($\theta$) | PSNR | SSIM | LPIPS | PSNR | SSIM | LPIPS | PSNR | SSIM | LPIPS |
| 30° | $36.84 \pm 0.14$ | 0.937 | 0.064 | $37.66 \pm 0.14$ | 0.944 | 0.068 | $38.04 \pm 0.15$ | 0.942 | 0.063 |
| 60° | $36.83 \pm 0.13$ | 0.937 | 0.064 | $37.65 \pm 0.15$ | 0.944 | 0.067 | $38.06 \pm 0.16$ | 0.942 | 0.063 |
| 90° | $36.84 \pm 0.07$ | 0.937 | 0.066 | $37.69 \pm 0.08$ | 0.945 | 0.069 | $38.07 \pm 0.09$ | 0.943 | 0.065 |
| 120° | $36.86 \pm 0.05$ | 0.937 | 0.067 | $37.69 \pm 0.07$ | 0.945 | 0.069 | $38.09 \pm 0.08$ | 0.943 | 0.065 |
| 150° | $36.79 \pm 0.20$ | 0.936 | 0.065 | $37.61 \pm 0.23$ | 0.944 | 0.069 | $38.00 \pm 0.24$ | 0.942 | 0.064 |
| 175° | $36.90 \pm 0.04$ | 0.938 | 0.065 | $37.74 \pm 0.03$ | 0.945 | 0.069 | $38.12 \pm 0.06$ | 0.943 | 0.064 |

### C.5.1 PRESERVATION OF PATHOLOGICAL STRUCTURES

A common concern with consistency-enforcing methods is the potential risk of over-smoothing, which might obscure critical pathological details such as tumors or lesions. It is crucial to distinguish the operating mechanism of ISCS from explicit smoothness priors like Total Variation (TV).

*Intuition* C.1 (Noise Consistency vs. Intensity Smoothness). Traditional priors (e.g., TV) operate directly on the signal intensity space $\mathcal{X}$, explicitly penalizing high-frequency gradients: $\min_{\mathbf{x}} \|\nabla \mathbf{x}\|_1$. This often leads to "cartoon-like" artifacts or the blurring of sharp boundaries, which can be detrimental for small lesions.

In contrast, ISCS operates in the *stochastic sampling trajectory* (noise space $\mathcal{Z}$). We enforce consistency on the noise $\mathbf{z}_t$, not on the pixel intensity $\mathbf{x}_0$. The structural details of the reconstruction are primarily determined by the measurement operator and the data consistency term. Consequently, ISCS encourages inter-slice coherence without suppressing the high-frequency signal components required to represent abrupt pathological changes.

**Validation on DeepLesion Dataset** We empirically validated this hypothesis using the DeepLesion dataset. We selected two representative subjects: Case A (1mm slice with a large lesion) and Case B (5mm slice with a small lesion). The visual results in Fig. 6 indicate:

- **Case A (Large Lesion, 1 mm):** ISCS successfully reconstructed the sharp boundaries and internal texture of the large tumor, demonstrating that Slerp interpolation in noise space does not blur structural transitions.

- **Case B (Small Lesion, 5 mm):** We observed that TV regularization, in its effort to enforce inter-slice intensity consistency, nearly obliterated the small tumor. In contrast, ISCS preserved the small lesion with high fidelity, yielding a result closest to the GT.

These results confirm that ISCS is robust in pathological scenarios, avoiding the detail destruction common in strong explicit smoothness regularizers.

### C.5.2 ROBUSTNESS TO VARYING SLICE THICKNESS

To evaluate the generalization of ISCS across different data acquisition protocols, we conducted experiments on datasets with varying slice thicknesses (non-uniform z-spacing), ranging from thin (3mm) to thick (7.5mm) slices. Specifically, we utilized subjects from the LDCT-PD dataset (3mm, 5mm) and the AbdomenAtlas1.0 dataset (7.5mm). Crucially, we applied the **exact same experimental settings and hyperparameters** across all cases, without any dataset-specific tuning.

The quantitative results in Table 9 show that ISCS consistently outperforms the DDS baseline across all metrics and axes. This robustness suggests that ISCS is a scalable solution for 3D reconstruction that does not require retraining or extensive hyperparameter tuning for different clinical protocols.

### C.6 ADDITIONAL VISUAL RESULTS

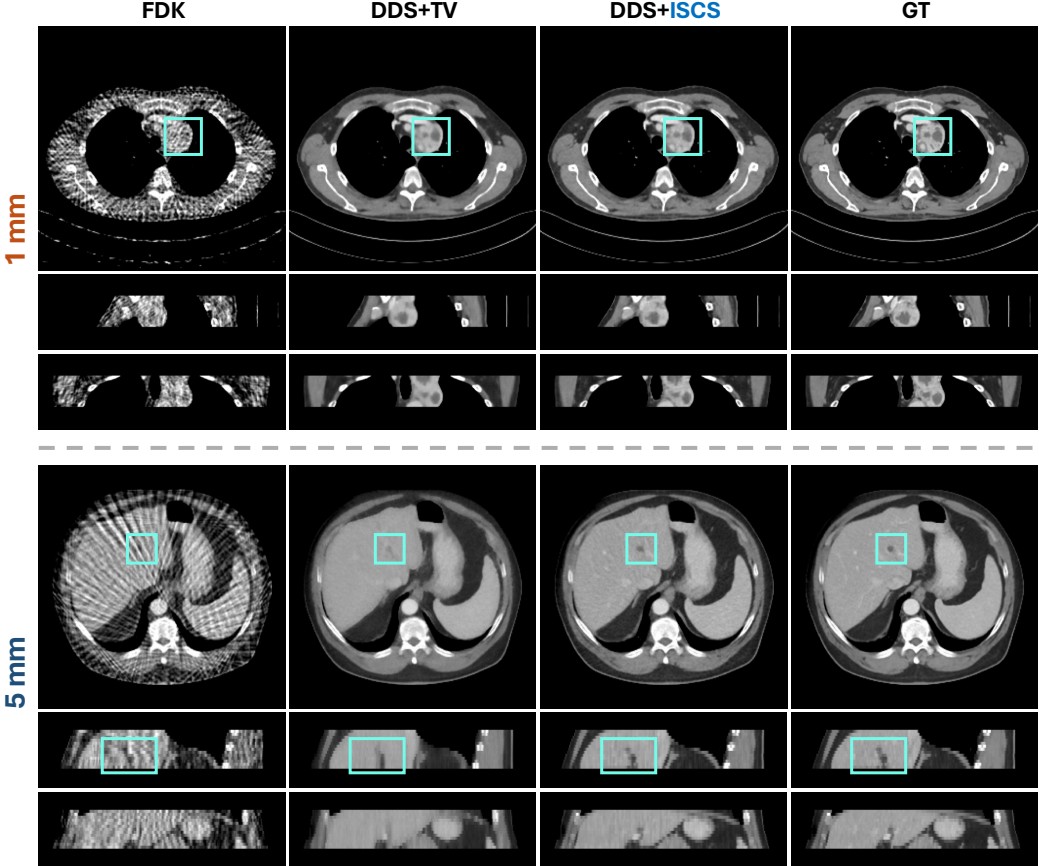

Figure 6: Qualitative comparison of compared methods for SVCT of 30 views. Two representative subjects with slice thicknesses of 1 mm (top) and 5 mm (bottom) are shown. Cyan boxes mark the lesion regions. The display window is [-175, 275] HU.

Table 9: **Quantitative Results under Varying Slice Thicknesses.** ISCS demonstrates consistent performance gains over the baseline (DDS) across all slice thicknesses (3mm, 5mm, 7.5mm). The method effectively mitigates inter-slice discontinuities regardless of z-spacing, confirming its robustness to variations in acquisition protocols.

| Thickness | Method | Axial | | | Coronal | | | Sagittal | | |
|---|---|---|---|---|---|---|---|---|---|---|
| | | PSNR | SSIM | LPIPS | PSNR | SSIM | LPIPS | PSNR | SSIM | LPIPS |
| 3mm | DDS | 36.17 | 0.945 | 0.046 | 37.06 | 0.932 | 0.082 | 36.97 | 0.932 | 0.073 |
| | DDS + ISCS | **37.48** | **0.951** | **0.049** | **38.54** | **0.956** | **0.044** | **38.45** | **0.956** | **0.038** |
| 5mm | DDS | 37.83 | 0.952 | 0.045 | 37.96 | 0.952 | 0.055 | 38.62 | 0.950 | 0.045 |
| | DDS + ISCS | **38.70** | **0.956** | **0.045** | **39.03** | **0.963** | **0.037** | **39.70** | **0.961** | **0.031** |
| 7.5mm | DDS | 34.97 | 0.934 | 0.043 | 35.34 | 0.938 | 0.027 | 37.15 | 0.940 | 0.026 |
| | DDS + ISCS | **36.02** | **0.941** | **0.040** | **36.68** | **0.950** | **0.023** | **38.14** | **0.951** | **0.021** |

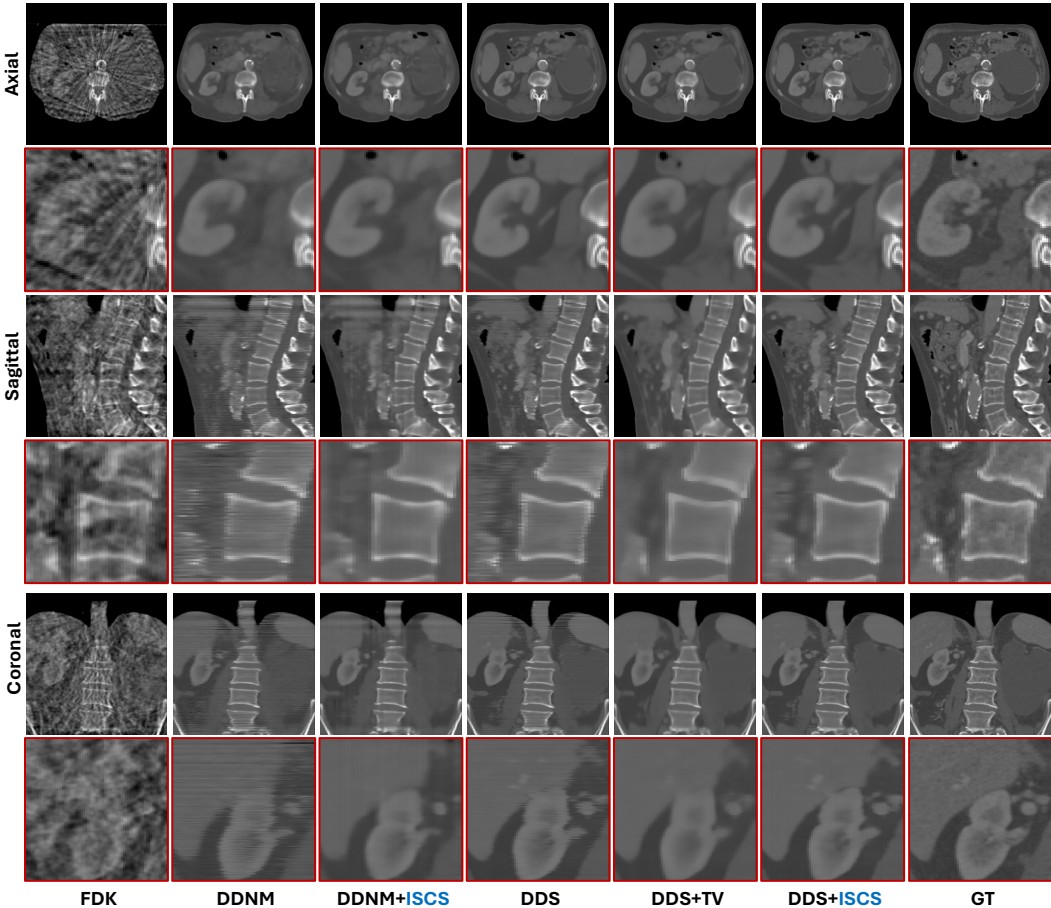

Figure 7: Qualitative results of compared methods on representative sample for SVCT of 30 views. The display window is set as [-480, 820] HU.

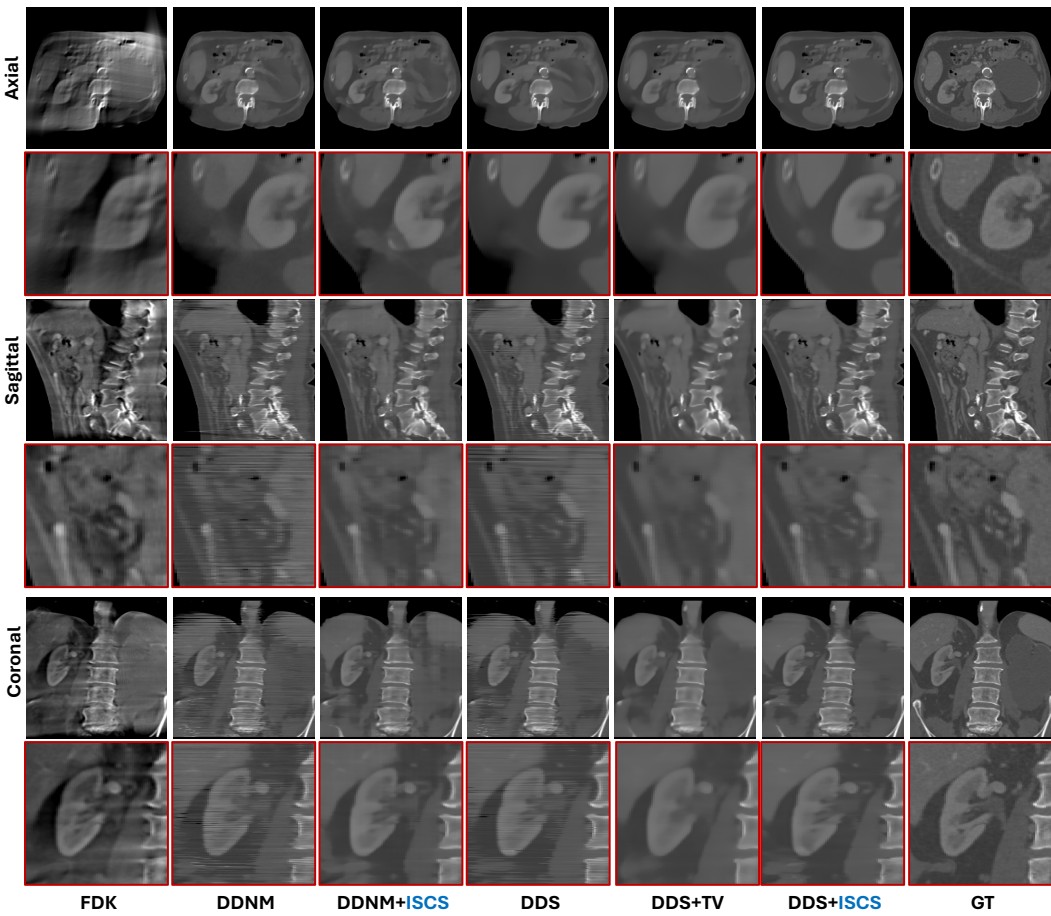

Figure 8: Qualitative results of compared methods on representative sample for LACT in $[0, 100°]$. The display window is set as [-480, 820] HU.

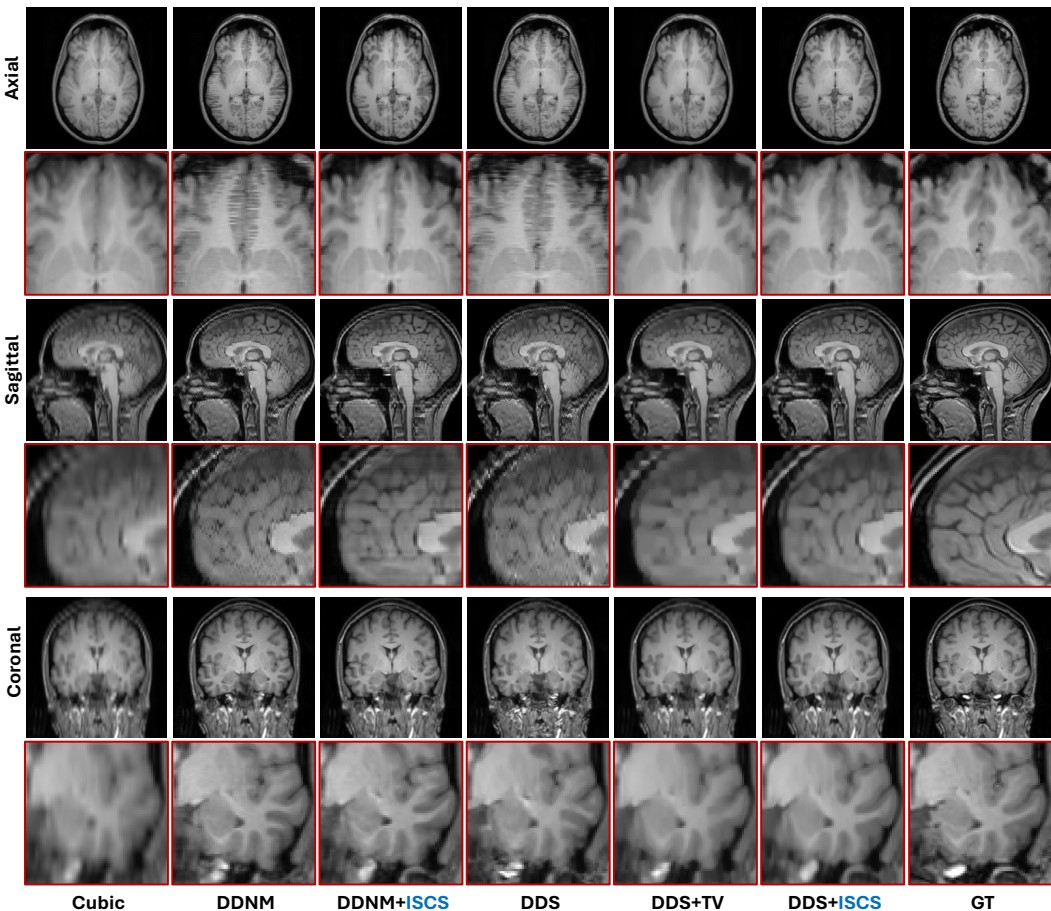

Figure 9: Qualitative results of compared methods on a representative sample for MRI SR of $5\times$.

