# — Supplementary Material —

# Improving 2D Diffusion Models for 3D Medical Imaging with Inter-Slice Consistent Stochasticity

**Table R 1.** Slice-to-slice difference of compared methods for three 3D medical imaging tasks: SVCT of 30 views, LACT of $[0, 100]°$, and MRI SR of $5\times$. $\text{SDiff}_{\text{recon}}$ and $\text{SDiff}_{\text{GT}}$ denote the mean absolute difference between adjacent slices for the reconstruction and ground truth, respectively; $\Delta = \text{SDiff}_{\text{recon}} - \text{SDiff}_{\text{GT}}$ measures the signed gap, and $|\Delta|$ is its absolute value (smaller is better).

| Task | Methods | Slice difference | | | |
|---|---|---|---|---|---|
| | | $\text{SDiff}_{\text{recon}}$ | $\text{SDiff}_{\text{GT}}$ | $\Delta$ | $|\Delta|$ |
| SVCT | FDK | 0.011893 | 0.006309 | 0.005584 | 0.005584 |
| | ADMM-TV | 0.004692 | 0.006309 | -0.001617 | 0.001617 |
| | DDNM | 0.015652 | 0.006309 | 0.009342 | 0.009342 |
| | DDNM+ISCS | 0.004524 | 0.006309 | -0.001785 | 0.001785 |
| | DDS | 0.011897 | 0.006309 | 0.005588 | 0.005588 |
| | DDS+TV | 0.003372 | 0.006309 | -0.002937 | 0.002937 |
| | DDS+ISCS | 0.004474 | 0.006309 | -0.001835 | 0.001835 |
| LACT | FDK | 0.007208 | 0.006309 | 0.000898 | 0.000898 |
| | ADMM-TV | 0.005432 | 0.006309 | -0.000877 | 0.000877 |
| | DDNM | 0.022753 | 0.006309 | 0.016443 | 0.016443 |
| | DDNM+ISCS | 0.004410 | 0.006309 | -0.001899 | 0.001899 |
| | DDS | 0.017901 | 0.006309 | 0.011592 | 0.011592 |
| | DDS+TV | 0.003743 | 0.006309 | -0.002566 | 0.002566 |
| | DDS+ISCS | 0.004344 | 0.006309 | -0.001966 | 0.001966 |
| MRI SR | Cubic | 0.007766 | 0.011934 | -0.004168 | 0.004168 |
| | ADMM-TV | 0.006861 | 0.011934 | -0.005074 | 0.005074 |
| | DDNM | 0.013414 | 0.011934 | 0.001480 | 0.001480 |
| | DDNM+ISCS | 0.010021 | 0.011934 | -0.001913 | 0.001913 |
| | DDS | 0.013787 | 0.011934 | 0.001853 | 0.001853 |
| | DDS+TV | 0.007202 | 0.011934 | -0.004732 | 0.004732 |
| | DDS+ISCS | 0.009838 | 0.011934 | -0.002096 | 0.002096 |

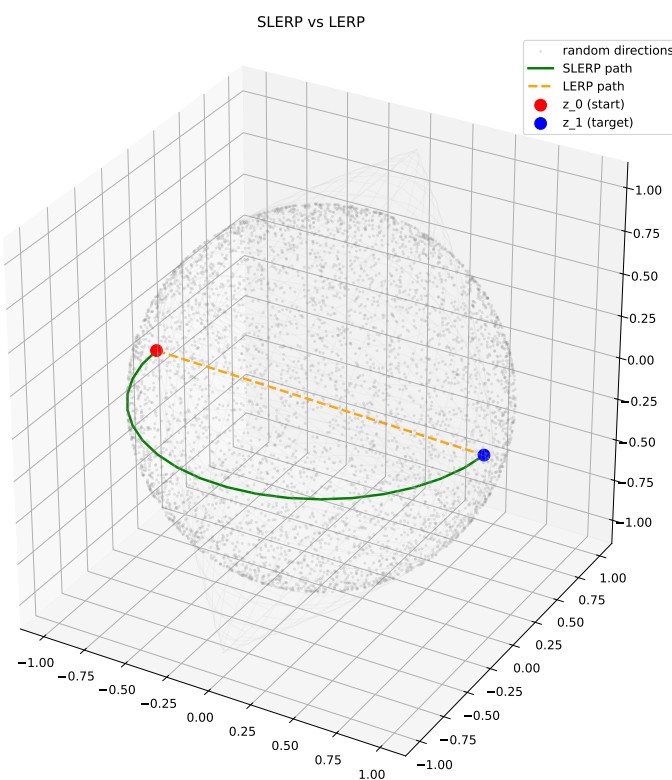

**Figure R 1.** Geometric interpretation of high-dimensional Gaussian noise and interpolation paths of SLERP and LERP.

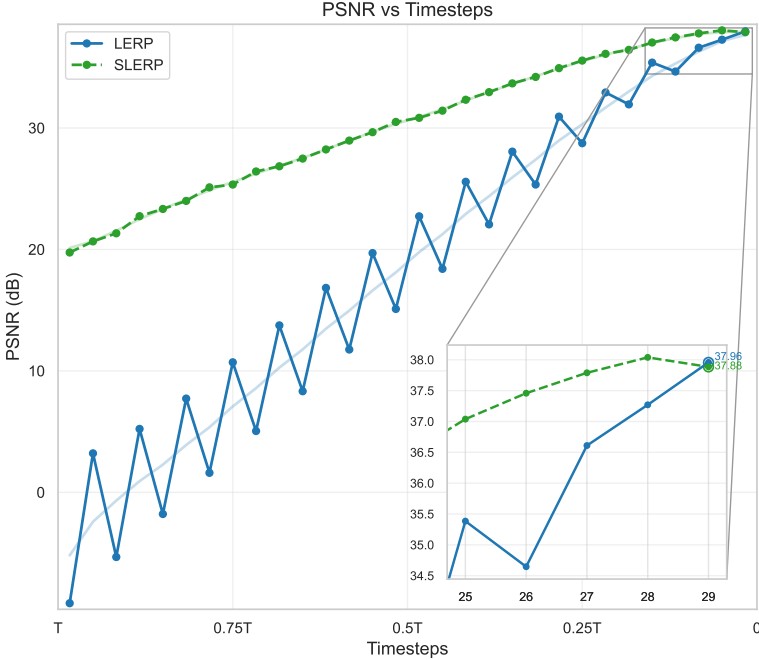

**Figure R 2.** Performance curves across the sampling process comparing LERP and SLERP as the inter-slice noise interpolation strategy in ISCS.

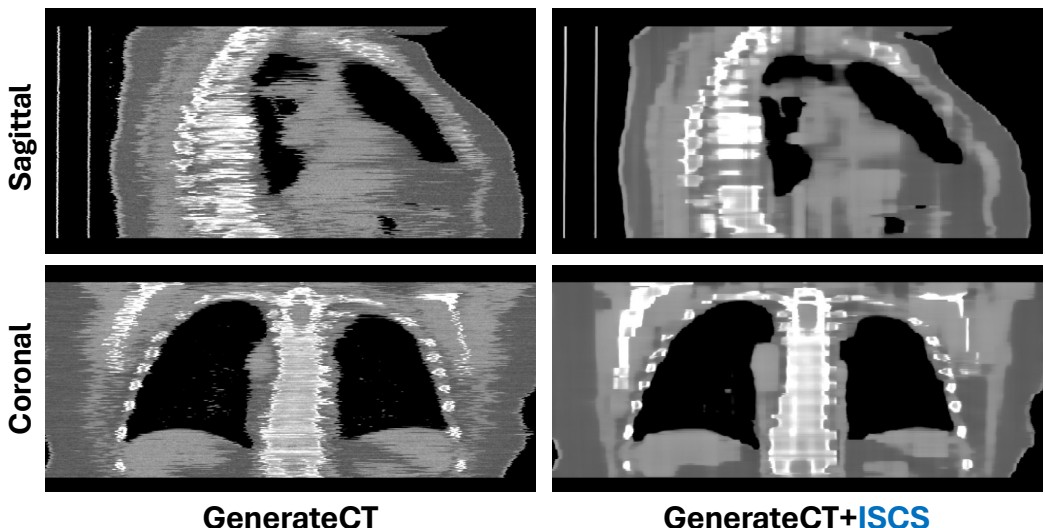

**Figure R 3.** Qualitative comparison of GenerateCT method with and without the proposed ISCS strategy.

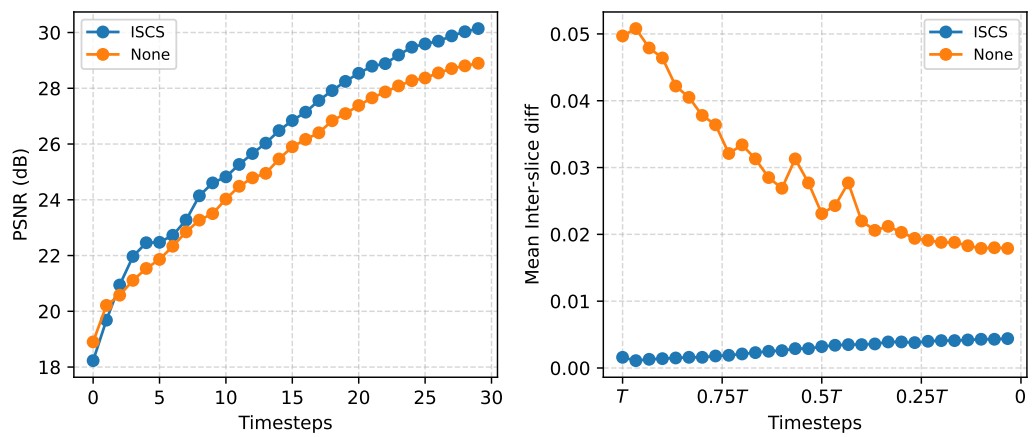

**Figure R 4.** Performance curves across the sampling proces of DDS with and without ISCS on LACT task of 0, 100]°.

162
163
164
165
166
167
168
169
170
171
172
173
174
175
176
177
178
179
180
181
182
183
184
185
186
187
188
189
190
191
192
193
194
195
196
197
198
199
200
201
202
203
204
205
206
207
208
209
210
211
212
213
214
215

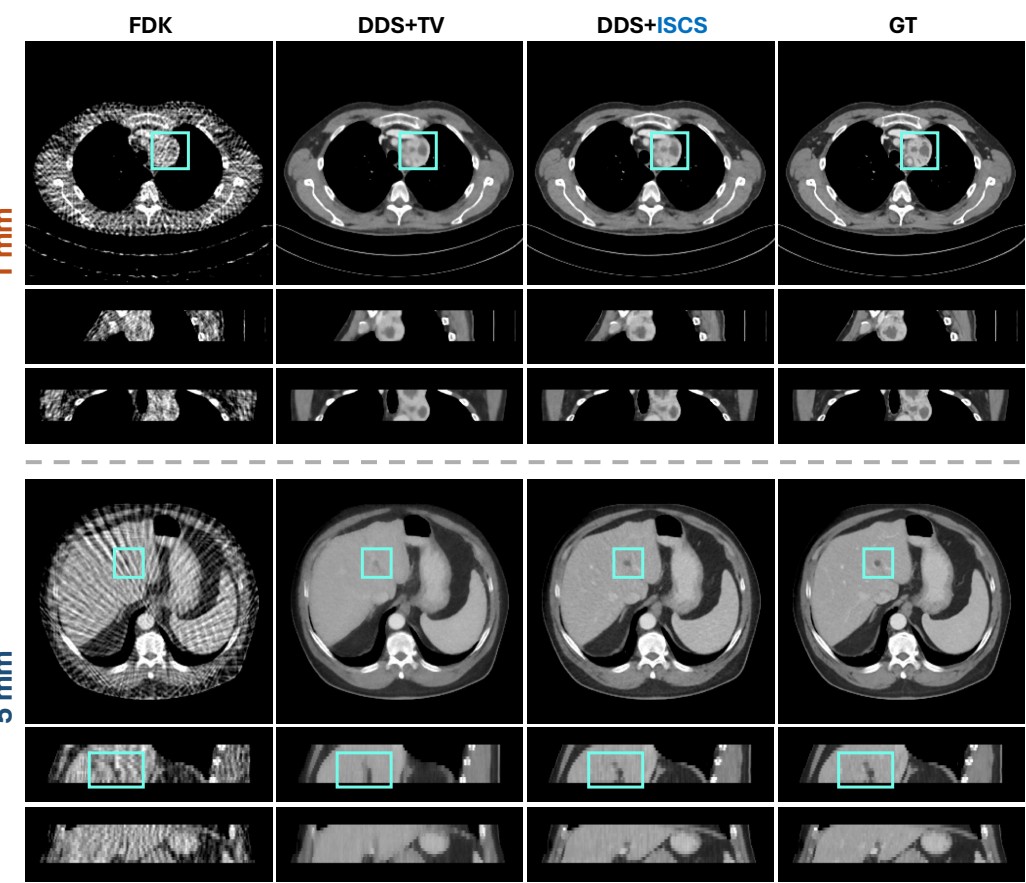

**Figure R 5.** Qualitative comparison of compared methods for SVCT of 30 views. Two representative subjects with slice thicknesses of 1 mm (top) and 5 mm (bottom) are shown. Cyan boxes mark the lesion regions. The display window is [-175, 275] HU.