# OpenReview forum: "Improving 2D Diffusion Models for 3D Medical Imaging with Inter‑Slice Consistent Stochasticity"
_ICLR.cc/2026/Conference — ICLR 2026 Poster_

### Official Review · Reviewer_6tkB · 2025-10-22

**Soundness:** 3
**Presentation:** 3
**Contribution:** 2
**Rating:** 4
**Confidence:** 4

**Summary:**

The authors exploit 2D diffusion models for 3D medical image reconstruction. The diffusion models are trained on xy-slices.
To obtain continuity in z-axis the authors apply correlated noise during sampling. They call this method "Inter-Slice Consistent Stochasticitiy" (ISCS). ISCS is an easy addition to the sampling pipeline, requiring only a single change in the algorithm.

**Strengths:**

- The paper is well-written and contains careful experiments. Importantly, according to Table 1 ISCS performs better than adding an additional TV-regulariser in the z-axis (which is currently the predominant way of enforcing inter-slice consistency)
- ISCS is a simple method (i.e., it contains no hyperparameters) which can be easily incorporated into existing code-bases with a few extra lines and comes with only negligible additional computational cost

Based on these points (better performance, simple implementation, no overhead computational cost) I would expect that this method will be well adapted in 3D-sampling tasks and code-bases.

**Weaknesses:**

- There are some important baselines missing in Table 1: using perpendicular 2d models [1] and batch-consistent sampling (i.e., using the same noise for all slices)
- There is no comparison against the deterministic DDIM sampler (i.e. setting $\eta=0$ in Equation (8)). For the case of deterministic samplers, no additional noise has to be added during sampling.
- With the adoption of flow-based models and deterministic samplers for diffusion models (e.g. PNDM [4]) the proposed method might no longer be relevant. For both flow-based models and deterministic sampler, the main argument for the lack of inter-slice consistency (starting at line 244) does no longer hold, because no noise is added.
- The authors provide a heuristic motivation why their method should work, but not really any theory. In contrast, using a TV-regulariser on the z-axis ensures consistency between slices (albeit requiring an additional hyperparameter)
- In line 264 the authors write "recent studies have demonstrated that random samples from a high-dimensional standard normal distribution tend to concentrate on the surface of a hypersphere" and cite two works from 2024. However, this is a pretty standard concentration of measure phenomenon and known for a long time.

[4] Liu et. al "Pseudo Numerical Methods for Diffusion Models in Manifolds" (2022)



[1] Lee et al. "Improving 3d imaging with pre-trained perpendicular 2d diffusion models"

**Questions:**

- How does the method depend on the sampling of the initial noise vectors for the first and the last slice?
- How does this method compare against the temporal correlated noise of [2] and the method in [3] ?

[2] Liu and Vahdat "On Equivariance and Fast Sampling in Video Diffusion Models Trained with Warped Noise" (2025)

[3] Chang et al. "How I warped your noise: a temporally correlated noise prior for diffusion models"

---

> ### Author Response · Authors · 2025-11-26
>
> We thank the reviewer for the thoughtful comments and for acknowledging our method as "simple," "effective," and likely to be "well adapted" by the community. We also appreciate the constructive criticism regarding baselines and theoretical justifications.
>
> ---
>
> **W-1. There are some important baselines missing in Table 1: using perpendicular 2d models [1] and batch-consistent sampling (i.e., using the same noise for all slices)**
>
> **A:** Batch-Consistent Sampling (BCS): We respectfully point out that we have already included the comparison with BCS (referred to as "Identical Noise" in our paper) in Section 4.3, Table 2, and Figure 4. The results demonstrate that while BCS ensures continuity, it causes severe "copying artifacts" (streaks) due to the rigid repetition of noise. ISCS effectively solves this by allowing smooth noise evolution. We will make this comparison more prominent in the revised main results.
>
> For the TPDM (perpendicular 2D models), see `Q-1 in the Global Responses` at the beginning of the rebuttal, please.
>
> ---
>
> **W-2. There is no comparison against the deterministic DDIM sampler (i.e. setting in Equation (8)). For the case of deterministic samplers, no additional noise has to be added during sampling.**
>
> **A:** See `Q-2 in the Global Responses` at the beginning of the rebuttal, please.
>
> ---
>
> **W-3. With the adoption of flow-based models and deterministic samplers for diffusion models (e.g. PNDM [4]) the proposed method might no longer be relevant. For both flow-based models and deterministic sampler, the main argument for the lack of inter-slice consistency (starting at line 244) does no longer hold, because no noise is added.**
>
> **A:** For deterministic samplers, our proposed ISCS remains applicable. See `Q-2 in the Global Responses` at the beginning of the rebuttal, please.
>
> ---
>
> **W-4. The authors provide a heuristic motivation why their method should work, but not really any theory. In contrast, using a TV-regulariser on the z-axis ensures consistency between slices (albeit requiring an additional hyperparameter).**
>
> **A:** We provide a **formal statistical analysis** and a **quantitative trajectory analysis** to elucidate the mechanism of ISCS.
>
> **(1) Theoretical Basis: Variance Reduction via Induced Correlation.** We can formalize the "variance reduction" mechanism using standard statistical theory applied to the inter-slice gradient. Let $x_i$ and $x_{i+1}$ be the reconstructed vectors of two adjacent slices. The inter-slice discontinuity can be measured by the variance of their difference:
> $$\text{Var}(x_i - x_{i+1}) = \text{Var}(x_i) + \text{Var}(x_{i+1}) - 2\text{Cov}(x_i, x_{i+1})$$
>
> * **Independent Sampling:** In standard slice-wise solvers, the stochastic noise is independent ($\text{Cov}(x_i, x_{i+1}) \approx 0$). The inter-slice variance is maximized, manifesting as high-frequency jitter along the z-axis.
> * **ISCS:** By utilizing Slerp to interpolate noise, ISCS explicitly introduces a strong positive correlation ($\text{Cov}(x_i, x_{i+1}) > 0$). Mathematically, this directly reduces the variance of the difference $\text{Var}(x_i - x_{i+1})$ without suppressing the signal variance $\text{Var}(x_i)$ (i.e., without blurring individual slices).
> * **Geometric Interpretation:** The inverse problem is ill-posed, meaning the measurement $y$ only constrains the solution to a subspace, leaving a "null-space" that diffusion priors must fill. Independent sampling fills this null-space randomly for each slice, causing divergence. ISCS effectively restricts the stochastic search space to a **smooth manifold tube**. This forces the "null-space filling" to be consistent across slices, thereby reducing the search variance.
>
> **(2) Quantitative Trajectory Analysis.** To quantitatively verify this, we tracked the evolution of the reconstruction metrics throughout the reverse diffusion timesteps. The results are visualized in `Fig. R 4 in the supplementary material`.
>
> * **Convergence Speed (Left Plot):** The PSNR trajectory of ISCS (blue) is consistently higher than that of independent sampling (orange) from the very early stages. This indicates that by narrowing the stochastic search space, ISCS prevents the solver from wasting iterations on inconsistent "jittery" solutions, thereby accelerating convergence toward the data manifold center (the Ground Truth).
> * **Stability (Right Plot):** The "Mean Inter-slice Difference" for ISCS remains extremely low and stable throughout the process. In contrast, the baseline struggles to reduce inter-slice differences effectively, exhibits non-monotonic fluctuations and settles at a significantly higher level of inconsistency due to the accumulation of uncorrelated stochasticity. This empirically confirms that ISCS enforces a stable convergence path where data consistency ($||y-Ax||^2$) and prior consistency work synergistically.

---

> > ### Author Response · Authors · 2025-11-26
> >
> > **W-5. In line 264 the authors write "recent studies have demonstrated that random samples from a high-dimensional standard normal distribution tend to concentrate on the surface of a hypersphere" and cite two works from 2024. However, this is a pretty standard concentration of measure phenomenon and known for a long time.**
> >
> > **A:** We agree that this is a standard phenomenon. We have revised the text to state it as a well-known property and adjusted the citations to refer to classical high-dimensional geometry literature [1] rather than framing it as a recent discovery.
> >
> > > [1] Vershynin, Roman. High-dimensional probability: An introduction with applications in data science. Vol. 47. Cambridge university press, 2018.
> >
> > ---
> >
> > **Q-1. How does the method depend on the sampling of the initial noise vectors for the first and the last slice?**
> >
> > **A:** See `Q-3 in the Global Responses` at the beginning of the rebuttal, please.
> >
> > ---
> >
> > **Q-2. How does this method compare against the temporal correlated noise of [2] and the method in [3] ?**
> >
> > **A:** We thank the reviewer for pointing out these advanced video diffusion methods. We have analyzed the applicability of the warping-based noise strategies proposed in [2] and [3] and compared our method with a representative video noise prior (PYoCo [1]).
> >
> > **(1) Methodological Difference.** The methods in [2] and [3] are excellent for video generation, but they rely on optical flow estimation to warp noise patterns, assuming that pixel changes correspond to object motion. In 3D medical imaging, the z-axis variation represents anatomical structural evolution, not temporal motion. While optical flow can theoretically model this as "deformable registration," estimating accurate deformation fields from sparse-view measurements (before reconstruction is complete) is an extremely ill-posed problem.
> >
> > **(2) Comparison with Video Noise Prior (PYoCo).** To provide a quantitative benchmark against video-domain techniques, we compared ISCS with PYoCo [1], a representative method for video noise correlation that is more adaptable to our setting than flow-based methods. As shown in `Table R1`, ISCS consistently outperforms PYoCo. For example, in the Sagittal view, ISCS achieves 38.21 dB PSNR, surpassing the best PYoCo result (37.66 dB).
> >
> >
> > |                          | **Axial** |           |           | **Coronal** |           |           | **Sagittal** |           |           |
> > | ------------------------ | --------- | --------- | --------- | ----------- | --------- | --------- | ------------ | --------- | --------- |
> > | **method**               | PSNR      | SSIM      | LPIPS     | PSNR        | SSIM      | LPIPS     | PSNR         | SSIM      | LPIPS     |
> > | **ISCS**                 | **36.98** | **0.938** | 0.063     | **37.80**   | **0.945** | 0.066     | **38.21**    | **0.943** | 0.061     |
> > | **PYoCo ($\alpha$=1)**   | 35.46     | 0.920     | 0.044     | 35.92       | 0.905     | 0.120     | 36.06        | 0.904     | 0.114     |
> > | **PYoCo ($\alpha$=3)**   | 36.29     | 0.928     | 0.041     | 36.99       | 0.926     | 0.075     | 37.22        | 0.924     | 0.070     |
> > | **PYoCo ($\alpha$=5)**   | 36.64     | 0.930     | **0.041** | 37.28       | 0.931     | **0.065** | 37.64        | 0.929     | 0.060     |
> > | **PYoCo ($\alpha$=7.5)** | 36.68     | 0.926     | 0.042     | 37.28       | 0.929     | 0.065     | 37.66        | 0.926     | **0.058** |
> > | **PYoCo ($\alpha$=10)**  | 36.42     | 0.923     | 0.043     | 37.02       | 0.925     | 0.067     | 37.31        | 0.923     | 0.061     |
> >
> > *Table R1. Comparison with the video noise prior method PYoCo [1] on SVCT-30 task.*
> >
> > **(3) Future Outlook.** We agree that the concept of "warping" from [2] and [3] is a valuable direction. A potential future extension of ISCS could involve a coarse-to-fine framework: generating a preliminary volume to estimate a deformation field, and then applying a "spatially warped ISCS" to guide the final sampling. While this increases complexity, it could combine the benefits of ISCS with anatomical guidance. We plan to explore this in future research.
> >
> > > [1] Ge, Songwei, et al. "Preserve your own correlation: A noise prior for video diffusion models." *ICCV 2023*.

---

> > > ### Comment · Reviewer_6tkB · 2025-11-27
> > >
> > > Thank you for the detailed reply and the additional experiments and clarifications.
> > >
> > > W-1: Missing Baselines
> > >
> > > I recognize that Table 2 includes a comparison to BCS. Still, the first thing a reader sees is the large Table 1 with an extensive set of baselines, and here the absence of BCS there stands out in my opinion.
> > > I also appreciate the comparison to TPDM and acknowledge that TPDM was designed for cubic volumes (256³).
> > >
> > > W-2 and W-3: Deterministic Samplers
> > >
> > > The comparison to deterministic samplers is useful. Is ISCS also used in  Table R4?
> > > Will these results (including with images of the reconstructions)appear in the revised manuscript?
> > >
> > > W-4: Theoretical Basis
> > >
> > > I am not convinced that the assumption of independent sampling holds, i.e., $\mathrm{Cov}(x_i, x_{i+1}) \approx 0$. The Data Fidelity Update (Line 143) acts on the full 3D volume, which should introduce correlations between neighboring slices.
> > >
> > > I also have a question about the Mean Inter-slice Difference (right plot, Fig. R.4 in the supplementary). You state:
> > >
> > > > “In contrast, the baseline exhibits a divergent trend in inter-slice differences as the stochasticity accumulates.”
> > >
> > > I don’t see a divergence in the baseline. The Mean Inter-slice Difference appears to decrease over iterations—just not to the same extent as ISCS.
> > >
> > > W-6: Comparison to Video Diffusion Techniques
> > >
> > > I was not familiar with PYoCo. Could you briefly discuss how it differs from ISCS? The Progressive Noise mode in PYoCo seems conceptually related.
> > > I agree that optical-flow-based approaches do not translate easily to medical imaging.

---

> > > > ### Author Response · Authors · 2025-11-27
> > > >
> > > > We thank the reviewer for the prompt feedback and the engaging discussion. We appreciate the recognition of our additional experiments. We address the follow-up questions below.
> > > >
> > > > ---
> > > >
> > > > **Regarding W-1**
> > > >
> > > > **A:** Comparison in Main Table We fully understand the reviewer’s perspective that Table 1 is the primary focus. However, after careful consideration, we have decided to retain the BCS comparison in the Ablation/Analysis section (Table 2) rather than moving it to Table 1, for the following reasons:
> > > >
> > > > * **Conceptual Categorization:** We view ISCS not merely as a new "solver," but as a methodology for "controlling stochasticity." In this context, BCS (Identical Noise) represents a specific, albeit sub-optimal, variant of stochasticity control. Therefore, comparing ISCS against BCS is methodologically an ablation study of noise strategies, whereas Table 1 focuses on comparing distinct inverse problem solvers (e.g., TV, DDNM, DDS).
> > > > * **Readability:** Table 1 is already dense with multiple baselines and metrics. Adding BCS might conflate the comparison between "solvers" and "noise definitions," potentially increasing the interpretation cost for readers.
> > > >
> > > > ---
> > > >
> > > > **Regarding W-2 & W-3**
> > > >
> > > > **A:** **Clarification:** We confirm that Table R4 was generated using the baseline sampler (DDS) without ISCS. **This was intended to demonstrate the general characteristic of DM-based solvers—that stochasticity is essential for inverse problem**.
> > > >
> > > > **Note on Metric Values:** The reviewer might notice that the quantitative values in Table R4 appear higher than those in Table 1. This is because this ablation was performed on a different subject than the one used for the main benchmarks. Additionally, these metrics were calculated on the full 3D volume (including large air regions), which tends to inflate values compared to tight-ROI evaluations.
> > > >
> > > > **Revision Plan:** We will include Table R3 (ISCS on $\eta=0$) and Table R4 (Ablation), along with the visual results, in the Appendix of the revised manuscript.
> > > >
> > > > ---
> > > >
> > > > **Regarding W-4**
> > > >
> > > > **A:** We appreciate this opportunity to refine our theoretical argument.
> > > >
> > > > * We agree that the Data Fidelity update ($||y-Ax||^2$) introduces inter-slice correlation. However, our point is that the Reverse Diffusion Sampling step acts as a counter-force. When noise is sampled independently, **the covariance of the injected noise** is effectively zero ($\text{Cov} \approx 0$) at that specific step.
> > > >
> > > > * Solution Space: Although the structure is partially correlated by $y$, the independent sampling significantly expands the feasible solution space, allowing adjacent slices to drift toward inconsistent local minima. ISCS restricts this "sampling solution space" by ensuring the noise itself has $\text{Cov} > 0$. By aligning the "noise correlation" with the "data correlation," ISCS creates a dual constraint that forces the solver to converge within a narrower, more consistent manifold.
> > > >
> > > > **Correction on "Divergence":** We apologize for the confusion caused by the term "divergent." We did not mean mathematical divergence to infinity. Rather, we meant that the baseline exhibits non-monotonic and unstable behavior in inter-slice differences during iterations, converging to a state with significantly higher inconsistency compared to the stable, monotonic improvement observed with ISCS. We have corrected this phrasing in the revision from
> > > > > "In contrast, the baseline exhibits a divergent trend in inter-slice differences as the stochasticity accumulates."
> > > >
> > > > to
> > > >
> > > > > "In contrast, the baseline struggles to reduce inter-slice differences effectively, exhibits non-monotonic fluctuations and settles at a significantly higher level of inconsistency due to the accumulation of uncorrelated stochasticity."
> > > >
> > > > ---

---

> > > > > ### Author Response · Authors · 2025-11-27
> > > > >
> > > > > **Regarding W-6**
> > > > >
> > > > > **A:** We thank the reviewer for asking this to clarify the distinction.
> > > > >
> > > > > **PYoCo Mechanism (Mixed Noise):** We choose the "Mixed Noise Model" in PYoCo (Ge et al., ICCV 2023), which blending a shared noise component (identical across frames) with an independent component as:
> > > > >   $$z_i = \frac{\alpha}{\sqrt{1 + \alpha^2}} z_{\text{shared}} + \frac{1}{\sqrt{1 + \alpha^2}} z_{\text{indep}, i}$$
> > > > > This approach creates a "static" correlation where every frame shares the same "base" noise. And as noted in our previous response, it is highly sensitive to the mixing ratio parameter $\alpha$.
> > > > >
> > > > > **ISCS Advantage (Continuous Transition):** In contrast, ISCS via Slerp creates a continuous geometric transition from the first slice to the last. This means that for any slice $i$, the noise is strongly and samely correlated with its immediate neighbors ($i \pm 1$) but the correlation naturally decays as the distance increases. Furthermore, the "direction" of correlation is distinct (interpolating from $z_1$ to $z_S$). This "evolving" noise trajectory matches the nature of medical volumes, where anatomy evolves continuously along the z-axis. PYoCo's "static mix" lacks this evolutionary property, making ISCS theoretically and empirically better suited for volumetric reconstruction.
> > > > >
> > > > > ---
> > > > >
> > > > > We once again thank the reviewer for their time and constructive comments. We hope that our response and additional experiments have addressed your questions and concerns. We would be happy to answer any further questions during the discussion period.

---

> > > > > ### Comment · Reviewer_6tkB · 2025-11-27
> > > > >
> > > > > **Regarding W-1**
> > > > >
> > > > > Given this argument, I feel that the comparison between DDS+BCS and DDS+ISCS (two different mechanisms for managing stochasticity) is a bit short, especially since it is based on a single dataset. Which experimental setting is actually used for this comparison, I could not find the information in Section 4.3
> > > > >
> > > > > Also, couldnt you argue that DDS+TV is also another form of stochasticity control? By enforcing smoothness, it effectively increases inter-slice correlation as well.
> > > > >
> > > > > On a different note: Figure 4 is a bit small, seeing the “copying artifacts” requires zooming in quite a lot. Maybe you can add a larger version of the figure?
> > > > >
> > > > > **Regarding W-4**
> > > > >
> > > > > What I had in mind was more of a formal mathematical result showing that ISCS provably increases correlation. That said, I do not think such a theorem is essential. Given that SLERP interpolation adds negligible overhead (compared to the full diffusion sampling pipeline) and ISCS consistently improves 3D sampling results over plain diffusion sampling (i.e. DDS vs DDS+ISCS), it feels already like a good default choice for practitioners.

---

> > > > > > ### Author Response · Authors · 2025-11-27
> > > > > >
> > > > > > We thank the reviewer for the prompt and constructive follow-up. We are encouraged by your assessment that ISCS is a "good default choice" for practitioners. We address your specific queries below.
> > > > > >
> > > > > > ---
> > > > > >
> > > > > > **Regarding W-1: Experimental Setting for BCS & Dataset Scope**
> > > > > >
> > > > > > **A:** **Experimental Setting:** We apologize for not detailing this in Section 4.3. The comparison in Table 2 was conducted on the L506 subject for the SVCT-30 task, averaged over 5 independent runs. To enable rapid evaluation for this specific ablation, we used a subset of 100 continuous slices rather than the full volume used in Table 1. This difference in evaluation volume (subset vs. full) explains the slight discrepancy in metric values between Table 1 and Table 2.
> > > > > >
> > > > > > **Expanded Evaluation:** We fully agree that a single dataset comparison is insufficient. We are currently running additional experiments to compare ISCS and BCS (Identical Noise) across more datasets and tasks, including full-volume CT reconstruction and MRI Super-Resolution. We will update the rebuttal and the revised manuscript with these comprehensive results as soon as they are ready.
> > > > > >
> > > > > > ---
> > > > > >
> > > > > > **Regarding W-1: TV as Stochasticity Control**
> > > > > >
> > > > > > **A:** **Conceptual Distinction:** We thank the reviewer for this insightful perspective. We agree that TV can be viewed as a regulator of stochasticity, but we wish to highlight a fundamental distinction in where this control is applied:
> > > > > >
> > > > > > * **TV (Outcome Control):** TV does not control the diffusion sampling process itself. Instead, it acts on the optimization outcome (the voxel intensities) by enforcing smoothness to penalize the stochastic variance manifesting in the final image. It manages the **symptoms** of randomness.
> > > > > > * **ISCS (Intrinsic Control):** In contrast, ISCS controls the intrinsic stochasticity of the sampling process (the noise source $\epsilon$). By correlating the random seed itself, it prevents the stochastic variance from **occurring in the first place**.
> > > > > >
> > > > > > This distinction (*`controlling the source (ISCS)` vs. `controlling the result (TV)`*) explains why ISCS avoids the over-smoothing trade-offs often inherent in TV regularization.
> > > > > >
> > > > > > ---
> > > > > >
> > > > > > **Regarding Figure 4: Visibility of Artifacts**
> > > > > >
> > > > > > **A:** We agree that the current figure is too small (due to initial layout constraints). We will enlarge Figure 4 in the revised manuscript. Additionally, we will add a new section in the Appendix containing more extensive qualitative comparisons (large-scale visualizations) between BCS and ISCS to clearly showcase the copying artifacts versus smooth transitions.
> > > > > >
> > > > > > ---
> > > > > >
> > > > > > **Regarding W-4: Formal Proof vs. Practical Utility**
> > > > > >
> > > > > > **A:** We sincerely thank the reviewer for their understanding. Deriving a formal mathematical proof for the convergence of non-linear diffusion posteriors is indeed challenging and likely beyond the scope of this paper. We hope that our provided statistical analysis (variance reduction) and quantitative trajectory analysis (empirical convergence plots) serve as sufficient proxies to justify the method's behavior. We are glad that the reviewer recognizes the practical value of ISCS as a robust "default choice."

---

### Official Review · Reviewer_fFZa · 2025-10-29

**Soundness:** 3
**Presentation:** 3
**Contribution:** 2
**Rating:** 2
**Confidence:** 4

**Summary:**

This paper addresses the issue of inter-slice inconsistency that arises when solving 3D medical inverse problems using 2D diffusion model-based inverse solvers (DIS). The authors propose a simple yet effective approach in which the noise used during the reverse diffusion denoising process is made slice-wise correlated. Motivated by recent findings showing that spherical linear interpolation (Slerp) is more meaningful than linear interpolation for Gaussian noise, the method samples two anchor noise vectors for the first and last slices, and then generates noise for intermediate slices via Slerp interpolation between these anchors.

**Strengths:**

1. The main strength of this work lies in its ability to mitigate inter-slice inconsistency—a major challenge in 3D medical inverse problems—without introducing additional priors or complex  modifications. Since the method only modifies the noise structure and introduces no new hyperparameters, it is highly practical and easy to integrate into existing 3D medical reconstruction pipelines.
2. Experiments across multiple imaging modalities and tasks provide strong evidence that the proposed method can effectively replace the widely used total variation (TV) prior.
3. The comparison with batch-consistent sampling (BCS) clearly demonstrates the limitations of BCS, specifically the copying artifacts it introduces, and shows that the proposed approach effectively resolves these issues.

**Weaknesses:**

1. The novelty of the proposed method is somewhat limited because similar structural noise techniques have been explored in video diffusion models for enforcing temporal consistency and reducing flickering artifacts (e.g., [1,2,3]). For instance, [1] also samples noise with inter-frame correlations via noise interpolation. To clearly distinguish the contribution, it is necessary to directly compare the proposed method with these noise prior approaches in the same manner as the comparison against BCS.
2. The interpolated noise is only used during the re-noising steps. Thus, when using deterministic DDIM sampling ($\eta=0$), where pure noise is not used, the method cannot be applied. If $\eta=0$ is rarely used in inverse problems due to degraded performance, the authors should provide supporting evidence or references. Additionally, since the proposed method depends directly on the degree of stochasticity ($\eta$), an analysis of how $\eta$ influences reconstruction quality is required.
3. BCS applies its procedure to the initial noise $x_T$. It seems natural to also apply the interpolated correlated noise to the $x_T$ in the proposed method. The authors should clarify why this was not done.
4. The angle (distance) between the two anchor noise vectors is randomly determined. If there exists an optimal anchor spacing for a given target volume, performance variance may increase. An ablation analyzing reconstruction quality with respect to anchor angle is needed.
5. **Minor:** The authors attribute the copying artifacts in medical volumes to domain differences from videos, but this claim lacks supporting evidence. Since prior BCS work does not show temporal-slice artifacts in video, the absence of such artifacts is not demonstrated. If the authors wish to maintain the domain-difference argument, they should show that BCS does *not* produce similar artifacts in video inverse problems; otherwise, the proposed method may be better positioned as a domain-genera**l** solution. Of course, evaluating video inverse problems may be beyond the scope of this paper — this is only a suggested clarification and does not affect the overall contribution.
6. **Minor:** Line 114: “A is invertible” appears to be a typo; presumably it should be **non-invertible**.

[1] Ge, Songwei, et al. "Preserve your own correlation: A noise prior for video diffusion models." *ICCV*. 2023.

[2] Wu, Tianxing, et al. "Freeinit: Bridging initialization gap in video diffusion models." *ECCV.* 2024.

[3] Chang, Pascal, et al. "How i warped your noise: a temporally-correlated noise prior for diffusion models." ICLR (2024).

**Questions:**

1. While the method shows that it can replace the widely used TV prior, it would be informative to see how it compares with 3D priors such as TQDM. It is okay to the proposed method does not outperform them.
2. Implicit slice-wise regularization via noise correlation may underperform explicit regularizers (e.g., TV) under severe degradation where large portions of information must be reconstructed. Does the method maintain performance under such challenging forward models?
3. Since the proposed approach seems orthogonal to explicit volumetric priors such as 3D TV, can the two be combined for further performance improvement?
4. If the above comparisons and ablation studies (as suggested in the weakness section) are included, the contribution would be significantly strengthened, and I would be willing to raise my score accordingly.

---

> ### Author Response · Authors · 2025-11-26
>
> We thank the reviewer for the detailed and thoughtful review, and for highlighting both the strengths and the limitations of our work. Below we address the main concerns point by point.
>
> ---
>
> **W-1. The novelty of the proposed method is somewhat limited because similar structural noise techniques have been explored in video diffusion models for enforcing temporal consistency and reducing flickering artifacts (e.g., [1,2,3]).**
>
> **A:** We sincerely thank the reviewer for pointing out the relevant literature in video diffusion. To clearly distinguish our contribution and demonstrate the domain-specific advantages of ISCS, we conducted a direct comparison with `PYoCo` [1], a representative noise correlation method from the video domain. We evaluated PYoCo on the SVCT-30 task. The results (reported in `Table R1`) show that ISCS outperforms PYoCo in terms of data fidelity (PSNR/SSIM) while maintaining competitive perceptual quality.
>
> ||**Axial**|||**Coronal**|||**Sagittal**|||
> |-|-|-|-|-|-|-|-|-|-|
> |**method**|PSNR|SSIM|LPIPS|PSNR|SSIM|LPIPS|PSNR|SSIM|LPIPS|
> |**ISCS**|**36.98**|**0.938**|0.063|**37.80**|**0.945**|0.066|**38.21**|**0.943**|0.061|
> |**PYoCo($\alpha$=1)**|35.46|0.920|0.044|35.92|0.905|0.120|36.06|0.904|0.114|
> |**PYoCo($\alpha$=3)**|36.29|0.928|0.041|36.99|0.926|0.075|37.22|0.924|0.070|
> |**PYoCo($\alpha$=5)**|36.64|0.930|**0.041**|37.28|0.931|**0.065**|37.64|0.929|0.060|
> |**PYoCo($\alpha$=7.5)**|36.68|0.926|0.042|37.28|0.929|0.065|37.66|0.926|**0.058**|
> |**PYoCo($\alpha$=10)**|36.42|0.923|0.043|37.02|0.925|0.067|37.31|0.923|0.061|
>
> *Table R1. Comparison with the video noise prior method PYoCo [1] on SVCT-30 task.*
>
> **Insight: Generation vs. Reconstruction.** Our experiments with PYoCo reveal a fundamental domain difference between video generation and medical inverse problems:
>
> - Diversity vs. Fidelity: Video generation typically aims for diversity and temporal dynamics, often favoring weaker noise correlations (e.g., $\alpha \approx 1$ in [1]) to allow for motion variance. In contrast, medical inverse problems aim to recover a specific, ground-truth volume from partial measurements, prioritizing fidelity and strict structural consistency.
> - Parameter Sensitivity: This distinction explains why PYoCo requires a much higher correlation strength ($\alpha=5$ to $7.5$) in our task compared to the video domain. Using the video default ($\alpha=1$) results in poor performance (PSNR ~35.46 dB) because it lacks the necessary "structural binding" to constrain the solution space effectively for reconstruction.
>
> **Trade-off and Future Outlook**
>
> - Flexibility vs. Stability: We acknowledge that methods like PYoCo offer flexibility through tunable parameters. However, this introduces an extra tuning cost to find the optimal hyperparameter for each new dataset or task. In the current context, ISCS demonstrates superior stability and efficiency, achieving optimal performance natively without the need for delicate parameter searching.
> - Broader Perspective: We sincerely thank the reviewer for bridging the connection to video diffusion research. While ISCS (Slerp) proves robust for our current scope, this perspective suggests that exploring more complex or learnable noise control mechanisms beyond simple interpolation is a valuable direction for future work to further enhance 2D-based medical reconstruction.
>
> **Contribution and Value.** While inspired by video methods, we believe our work holds distinct value for the medical community. *To the best of our knowledge, we are the first to introduce the perspective of "stochasticity control" to 2D DM-based medical inverse problems*. Unlike the video domain, where large-scale 3D/temporal datasets are abundant, high-quality 3D medical data is scarce and privacy restricted. Training 3D medical diffusion models is exceptionally difficult. Therefore, our method, which empowers easily trainable 2D models to achieve 3D consistency without retraining, addresses a specific pain point in medical imaging that is distinct from the video generation context.
>
> > [1] Ge, Songwei, et al. "Preserve your own correlation: A noise prior for video diffusion models." ICCV. 2023.
>
> ---
>
> **W-2. The interpolated noise is only used during the re-noising steps. Thus, when using deterministic DDIM sampling (eta=0), where pure noise is not used, the method cannot be applied. If  is rarely used in inverse problems due to degraded performance, the authors should provide supporting evidence or references. Additionally, since the proposed method depends directly on the degree of stochasticity (eta=0), an analysis of how influences reconstruction quality is required.**
>
> **A:** See `Q-2 in the Global Responses` at the beginning of the rebuttal, please.

---

> > ### Author Response · Authors · 2025-11-26
> >
> > **W-3. BCS applies its procedure to the initial noise. It seems natural to also apply the interpolated correlated noise to the in the proposed method. The authors should clarify why this was not done.**
> >
> > **A:** We apologize for the confusion caused by the description. We clarify that **we indeed apply ISCS to the initial noise $x_T$ in our implementation**. The initial noise volume is generated by interpolating between two anchor noise vectors using Slerp. This ensures that the starting point of the reverse process is already coherent across slices. We have updated the Method section (Algorithm 1) to explicitly state this step.
> >
> > ---
> >
> > **W-4. The angle (distance) between the two anchor noise vectors is randomly determined. If there exists an optimal anchor spacing for a given target volume, performance variance may increase. An ablation analyzing reconstruction quality with respect to anchor angle is needed.**
> >
> > **A:** We first clarify that in high-dimensional spaces (e.g., $D = 256 \times 256$), the "randomness" of the angle is counter-intuitively stable. Due to the concentration of measure phenomenon, the angle between any two independently sampled standard normal vectors concentrates tightly around 90 degrees (orthogonality) with negligible variance. Therefore, our default random initialization essentially guarantees a constant interpolation arc length, minimizing performance variance. We have addressed this through both theoretical clarification and a comprehensive ablation study. **See `Q-3 in the Global Responses` at the beginning of the rebuttal, please.**
> >
> > ---
> >
> > **W-5. Minor: The authors attribute the copying artifacts in medical volumes to domain differences from videos, but this claim lacks supporting evidence. Since prior BCS work does not show temporal-slice artifacts in video, the absence of such artifacts is not demonstrated. If the authors wish to maintain the domain-difference argument, they should show that BCS does not produce similar artifacts in video inverse problems; otherwise, the proposed method may be better positioned as a domain-general solution. Of course, evaluating video inverse problems may be beyond the scope of this paper — this is only a suggested clarification and does not affect the overall contribution.**
> >
> > **A:** We thank the reviewer for this logical scrutiny. We agree that verifying the behavior on video data is beyond the scope of this work. Instead of claiming a fundamental domain incompatibility, we clarify our argument based on the difference in perception and structural continuity:
> >
> > - **Perceptual Difference:** In video generation, identical noise (BCS) creates a "frozen noise" effect. Since videos are viewed temporally, this often results in a stable background, which is perceptually acceptable or even desirable for stability.
> > - **Structural Manifestation in 3D:** In medical imaging, the "temporal axis" is actually the spatial Z-axis. When we reconstruct a 3D volume, we visualize it from orthogonal views (Coronal/Sagittal). In this context, the "frozen noise" of BCS manifests physically as longitudinal streaks (copying artifacts) that cut through the anatomy. What might be a "stable background" in a video becomes an "unnatural artifact" in a 3D medical volume because anatomical structures evolve continuously along the Z-axis and should not have identical high-frequency noise patterns drilled through them.
> > - **Positioning:** We appreciate the reviewer's suggestion. While we believe this phenomenon is particularly detrimental in medical 3D reconstruction due to the need for multi-planar analysis, we are open to the possibility that ISCS could also serve as a domain-general solution to alleviate "texture sticking" in videos, although confirming this is left for future work.
> >
> > ---
> >
> > **W-6. Minor: Line 114: “A is invertible” appears to be a typo; presumably it should be non-invertible.**
> >
> > **A:** We thank the reviewer for catching the typo on line 114. We will correct “A is invertible” to “A is non invertible” in the revised manuscript.
> >
> > ---
> >
> > **Q-1. While the method shows that it can replace the widely used TV prior, it would be informative to see how it compares with 3D priors such as TQDM. It is okay to the proposed method does not outperform them.**
> >
> > **A:** See `Q-1 in the Global Responses` at the beginning of the rebuttal, please.

---

> > > ### Author Response · Authors · 2025-11-26
> > >
> > > **Q-2. Implicit slice-wise regularization via noise correlation may underperform explicit regularizers (e.g., TV) under severe degradation where large portions of information must be reconstructed. Does the method maintain performance under such challenging forward models?**
> > >
> > > **A:** We conducted a "stress test" on the SVCT task with extremely sparse views (4, 8, and 20 views). The results (presented in `Table R2`) reveal an interesting trade-off between explicit intensity smoothing (TV) and implicit stochastic consistency (ISCS).
> > >
> > > - **Extreme Degradation (4 & 8 views):** Explicit TV regularization outperforms ISCS in PSNR and SSIM by enforcing strong intensity smoothness, which effectively "inpaints" large missing areas with smooth gradients. However, ISCS consistently achieves better LPIPS scores, indicating that even in these extreme cases, ISCS preserves better perceptual texture quality and avoids the "cartoon-like" over-smoothing typical of TV.
> > > - **Moderate Degradation (20 views):** As the view number increases to 20 (still a challenging sparse scenario), ISCS surpasses TV across all metrics (PSNR, SSIM, and LPIPS). This demonstrates that as long as sufficient structural information exists, ISCS recovers fine details and 3D consistency better than explicit regularization.
> > >
> > > While TV obtain slightly better quantitative results in extremely ill-posed regimes (4/8 views), we note that such extreme degradation typically yields clinically unusable results regardless of the method, arguably falling outside the effective scope of unsupervised diffusion solvers. However, in the workable regime (e.g., $\ge 20$ views), ISCS proves to be the superior choice for high-fidelity reconstruction.
> > >
> > > |Thickness|Method|Axial|||Coronal|||Sagittal|||
> > > |-|-|-|-|-|-|-|-|-|-|-|
> > > |||PSNR|SSIM|LPIPS|PSNR|SSIM|LPIPS|PSNR|SSIM|LPIPS|
> > > |*4 views*|DDS|22.51|0.662|0.162|22.31|0.550|0.382|22.33|0.543|0.383|
> > > ||DDS+TV|**25.34**|**0.741**|0.164|**25.58**|**0.785**|0.200|**25.83**|**0.777**|0.184|
> > > ||DDS+ISCS|23.89|0.704|**0.156**|24.17|0.744|**0.190**|24.28|0.735|**0.180**|
> > > |*8 views*|DDS|26.04|0.756|0.131|25.84|0.686|0.328|25.86|0.683|0.324|
> > > ||DDS+TV|**28.98**|**0.827**|0.145|**29.60**|**0.853**|0.200|**29.90**|**0.846**|0.184|
> > > ||DDS+ISCS|28.36|0.810|**0.123**|28.94|0.836|**0.134**|29.14|0.829|**0.124**|
> > > |*20 views*|DDS|33.63|0.903|0.059|33.78|0.880|0.144|33.97|0.878|0.137|
> > > ||DDS+TV|34.78|0.896|0.058|35.47|0.901|0.094|35.69|0.899|0.086|
> > > ||DDS+ISCS|**35.47**|**0.917**|**0.057**|**36.28**|**0.925**|**0.066**|**36.47**|**0.922**|**0.060**|
> > >
> > > *Table R2. Comparison between ISCS and TV under extremely sparse-view CT reconstruction tasks.*
> > >
> > > ---
> > >
> > > **Q-3. Since the proposed approach seems orthogonal to explicit volumetric priors such as 3D TV, can the two be combined for further performance improvement?**
> > >
> > > **A:** We thank the reviewer for this insightful suggestion. We agree that ISCS (which regulates stochasticity) and 3D TV (which regulates intensity smoothness) operate on orthogonal mechanisms and can theoretically be complementary.
> > >
> > > To verify this, we conducted a preliminary experiment on the SVCT-30 task. We applied a very weak TV regularization on top of our standard DDS+ISCS pipeline. The quantitative results are reported in `Table R3`.
> > >
> > > The combination (DDS+ISCS+TV) achieves higher PSNR and significantly improved LPIPS (e.g., Axial LPIPS drops from 0.063 to 0.039), confirming that explicit volumetric priors can further suppress noise and improve perceptual quality when combined with ISCS. Notably, because ISCS already establishes a strong baseline consistency, we only needed to apply a minimal TV weight. This largely avoids the severe "cartoon-like" or "staircase" artifacts typically associated with strong TV regularization, although a slight drop in SSIM suggests some residual smoothing of fine textures.
> > >
> > > While effective, introducing TV inevitably brings the burden of hyperparameter tuning (e.g., weight selection), contrasting with the parameter-free nature of ISCS. Therefore, while we demonstrate the feasibility of this combination here, we believe the most promising future direction lies in combining ISCS with more advanced learned volumetric priors or shape priors, rather than simple handcrafted regularizers.
> > >
> > > |              |       | DDS+ISCS  | DDS+ISCS+TV |
> > > | ------------ | ----- | --------- | ----------- |
> > > | **Axial**    | PSNR  | 36.98     | **37.23**   |
> > > |              | SSIM  | **0.938** | 0.932       |
> > > |              | LPIPS | 0.063     | **0.039**   |
> > > | **Coronal**  | PSNR  | 37.80     | **38.00**   |
> > > |              | SSIM  | **0.945** | 0.938       |
> > > |              | LPIPS | 0.066     | **0.059**   |
> > > | **Sagittal** | PSNR  | 38.21     | **38.33**   |
> > > |              | SSIM  | **0.943** | 0.936       |
> > > |              | LPIPS | 0.061     | **0.053**   |
> > >
> > > *Table R3. Combining ISCS with TV regularization on SVCT-30 task.*

---

> ### Comment · Reviewer_fFZa · 2025-11-27
>
> The authors’ rebuttal and additional experiments have clearly addressed all of my concerns. I am particularly impressed that every issue I raised was resolved both experimentally and through careful reasoning, and I appreciate the authors’ thorough response. I also look forward to seeing this approach broadly adopted in future work on 3D medical inverse problems. As there remain no substantive concerns that would justify rejecting the paper, I am significantly increasing my score.

---

> > ### Author Response · Authors · 2025-11-27
> > **Sincere Gratitude for Your Reassessment**
> >
> > We are absolutely thrilled and deeply grateful for your positive reassessment and the significant score increase!
> >
> > We are heartened to learn that our additional experiments and reasoning have satisfactorily resolved all your concerns. Your constructive feedback, **particularly regarding the comparison with video diffusion models and the clarification of domain-specific challenges**, pushed us to dig deeper and has substantially strengthened the quality and rigor of our work.
> >
> > Thank you once again for your time and invaluable guidance. Wish you a wonderful day!

---

### Official Review · Reviewer_PqvB · 2025-10-31

**Soundness:** 4
**Presentation:** 4
**Contribution:** 3
**Rating:** 6
**Confidence:** 5

**Summary:**

This paper addresses a key limitation of using 2D diffusion models (DMs) for 3D medical imaging: stochastic inconsistencies between independently sampled slices, leading to discontinuous 3D volumes. The authors propose Inter-Slice Consistent Stochasticity (ISCS)---a plug-and-play method that enforces inter-slice correlation in the diffusion sampling stage by introducing spherical linear interpolation (Slerp) between noise vectors, ensuring smooth stochastic transitions across slices. The method requires no retraining, extra loss terms, or additional computational cost, and can be integrated into existing diffusion-based inverse solvers (DDNM, DDS). Experiments on sparse-view CT, limited-angle CT, and MRI isotropic SR demonstrate consistent PSNR/SSIM/LPIPS improvements, outperforming TV-regularized baselines.

**Strengths:**

1. The paper clearly identifies uncorrelated stochasticity as the fundamental cause of inter-slice artifacts, a well-reasoned observation extending findings from video DMs (Kwon & Ye, 2025) to 3D medical imaging.

2. ISCS is lightweight and does not modify model training or inference pipelines. The Slerp-based correlated noise interpolation is mathematically sound and geometrically intuitive.

3. Strong empirical validations:
- Demonstrated across three inverse problems (SVCT, LACT, MRI-SR).
- Improves both fidelity (PSNR/SSIM) and perceptual quality (LPIPS).
- Outperforms both baseline DIS (DDNM/DDS) and TV-regularized counterparts, maintaining anatomical details without oversmoothing.

4. The plug-and-play property enables easy adoption in other 2D DM-based 3D reconstruction pipelines.

**Weaknesses:**

1. While geometrically motivated, there is no formal proof or quantitative analysis on how inter-slice correlation affects posterior sampling convergence or variance reduction.

2. Scalability and generalization:
- The Slerp interpolation assumes smooth anatomical transitions, which may fail in pathological cases (e.g., tumors, lesions) with abrupt structure changes.
- Performance under varying slice thickness or non-uniform z-spacing is not evaluated.

3. The paper lacks comparison with recent 3D-aware diffusion priors (e.g., DiffusionBlend [Song et al., NeurIPS '24]) or multi-plane consistency methods.

4. Only Slerp vs. identical noise is analyzed. Additional ablations (e.g., interpolation degree, anchor distance, or adaptive correlation strength) would strengthen claims.

**Questions:**

1. How does inter-slice correlation affect posterior sampling convergence or variance reduction? I expect formal proof or (at minimum) quantitative analysis.

2. How does ISCS interact with deterministic samplers (e.g., DDIM with $\eta = 0$)?

3. How does Slerp interpolation work if abrupt structure changes exist?

4. I strongly suggest including comparisons with recent 3D-aware diffusion priors (e.g., DiffusionBlend [Song et al., NeurIPS '24]) or multi-plane consistency methods.

5. Could Slerp interpolation be extended to spatially adaptive correlation where anatomical gradient drives $\alpha_i$?

6. How does ISCS perform with very thick slices or anisotropic resolutions?

7. Are there any cases where ISCS causes artifacts or over-correlation (e.g., repetitive patterns)?

8. Is the improvement consistent across random seeds, or does stochastic correlation introduce bias?

---

> ### Author Response · Authors · 2025-11-26
>
> We thank the reviewer for the careful reading and the constructive suggestions. Below we respond to the main concerns.
>
> ---
>
> **W-1 & Q-1: Analysis on Convergence and Variance Reduction via ISCS**
>
> **A:** We thank the reviewer for this profound question. While a rigorous derivation of the convergence bounds for non-linear diffusion posteriors is mathematically intractable and remains an open problem in the field, we provide a **formal statistical analysis** and a **quantitative trajectory analysis** to elucidate the mechanism of ISCS.
>
> **(1) Theoretical Basis: Variance Reduction via Induced Correlation.** We can formalize the "variance reduction" mechanism using standard statistical theory applied to the inter-slice gradient. Let $x_i$ and $x_{i+1}$ be the reconstructed vectors of two adjacent slices. The inter-slice discontinuity can be measured by the variance of their difference:
> $$\text{Var}(x_i - x_{i+1}) = \text{Var}(x_i) + \text{Var}(x_{i+1}) - 2\text{Cov}(x_i, x_{i+1})$$
> * **Independent Sampling:** In standard slice-wise solvers, the stochastic noise is independent ($\text{Cov}(x_i, x_{i+1}) \approx 0$). The inter-slice variance is maximized, manifesting as high-frequency jitter along the z-axis.
> * **ISCS:** By utilizing Slerp to interpolate noise, ISCS explicitly introduces a strong positive correlation ($\text{Cov}(x_i, x_{i+1}) > 0$). Mathematically, this directly reduces the variance of the difference $\text{Var}(x_i - x_{i+1})$ without suppressing the signal variance $\text{Var}(x_i)$ (i.e., without blurring individual slices).
> * **Geometric Interpretation:** The inverse problem is ill-posed, meaning the measurement $y$ only constrains the solution to a subspace, leaving a "null-space" that diffusion priors must fill. Independent sampling fills this null-space randomly for each slice, causing divergence. ISCS effectively restricts the stochastic search space to a **smooth manifold tube**. This forces the "null-space filling" to be consistent across slices, thereby reducing the search variance.
>
> **(2) Quantitative Trajectory Analysis.** To quantitatively verify this, we tracked the evolution of the reconstruction metrics throughout the reverse diffusion timesteps. The results are visualized in `Fig. R 4 in the supplementary material`.
>
> *  The PSNR trajectory of ISCS (blue) is consistently higher than that of independent sampling (orange) from the very early stages. This indicates that by narrowing the stochastic search space, ISCS prevents the solver from wasting iterations on inconsistent "jittery" solutions, thereby accelerating convergence toward the data manifold center (the Ground Truth).
> *  The "Mean Inter-slice Difference" for ISCS remains low and stable throughout the process. In contrast, the baseline struggles to reduce inter-slice differences effectively, settling at a significantly higher level of inconsistency due to the accumulation of uncorrelated stochasticity. This empirically confirms that ISCS enforces a stable convergence path where data consistency ($||y-Ax||^2$) and prior consistency work synergistically.
>
> ---
>
> **W-2. Scalability and generalization:**
>
> **W-2 (a) & Q-3. Pathological cases (e.g., tumors, lesions) with abrupt structure changes**
>
> **A:** ISCS operates fundamentally differently from explicit smoothness priors like TV. ISCS enforces consistency on the stochastic sampling trajectories (the noise), not on the image intensity; Unlike TV regularization, which explicitly penalizes intensity gradients (often leading to "cartoon-like" artifacts or blurring of sharp boundaries). In ISCS, the actual structures in the reconstruction are still primarily determined by the measurement operator and the data consistency term. This is also supported by the visual results in Fig. 2 and 3 in the submission manuscript, where ISCS preserves thin structures and subtle textures better than TV based baselines.
>
> **Validation on DeepLesion Dataset:** To empirically validate this on pathological cases, we conducted a new experiment using the DeepLesion dataset. We selected two representative subjects: (A) a thin-slice volume (1mm) containing a large lesion, and (B) a thick-slice volume (5mm) containing a small lesion. The results are visualized in `Fig. R 5 in the supplementary material`.
>
> * Case A (1mm, Large Lesion): ISCS successfully reconstructed the sharp boundaries of the large tumor, demonstrating that Slerp does not blur structural transitions.
>
> * Case B (5mm, Small Lesion): We observed that TV regularization, in its effort to enforce inter-slice intensity consistency, nearly obliterated the small tumor, rendering it indistinguishable. In contrast, ISCS preserved this with high fidelity, yielding a result closest to the GT.
>
> These results suggest that although ISCS encourages inter-slice coherence, it does not destroy pathological details in the way that strong explicit smoothness regularizers can, and remains robust in both thin-slice and thick-slice pathological scenarios.

---

> > ### Author Response · Authors · 2025-11-26
> >
> > **W-2 (b) & Q-6. Performance under varying slice thickness or non-uniform z-spacing is not evaluated**
> >
> > **A:** We have addressed this through both extensive empirical validation and a theoretical discussion on adaptive interpolation.
> >
> > * **Empirical Validation on Varying Thicknesses:** To evaluate the robustness of ISCS, we conducted additional experiments on CT reconstruction tasks involving datasets with varying slice thicknesses, ranging from thin to thick slices. Specifically:
> >   * **Datasets:** We selected subjects with **3mm** and **5mm** slice thicknesses from the LDCT-PD dataset, and a subject with **7.5mm** slice thickness from the AbdomenAtlas1.0 dataset.
> >   * **Robustness:** Crucially, we applied the exact same experimental settings and hyperparameters across all cases, without any dataset-specific tuning.
> >   * As shown in the `Tab. R1`, ISCS demonstrates consistent performance gains across all slice thicknesses. It effectively mitigates inter-slice discontinuities and improves 3D coherence regardless of the z-spacing, confirming that our method is robust to variations in data acquisition protocols.
> >
> > |Thickness|Method|Axial|||Coronal|||Sagittal|||
> > |-|-|-|-|-|-|-|-|-|-|-|
> > |||PSNR|SSIM|LPIPS|PSNR|SSIM|LPIPS|PSNR|SSIM|LPIPS|
> > |3mm|DDS|36.17|0.945|0.046|37.06|0.932|0.082|36.97|0.932|0.073|
> > ||DDS+ISCS|**37.48**|**0.951**|**0.049**|**38.54**|**0.956**|**0.044**|**38.45**|**0.956**|**0.038**|
> > |5mm|DDS|37.83|0.952|0.045|37.96|0.952|0.055|38.62|0.950|0.045|
> > ||DDS+ISCS|**38.70**|**0.956**|**0.045**|**39.03**|**0.963**|**0.037**|**39.70**|**0.961**|**0.031**|
> > |7.5mm|DDS|34.97|0.934|0.043|35.34|0.938|0.027|37.15|0.940|0.026|
> > ||DDS+ISCS|**36.02**|**0.941**|**0.04**|**36.68**|**0.95**|**0.023**|**38.14**|**0.951**|**0.021**|
> >
> > *Table R1. Quantitative results of ISCS on CT reconstruction tasks with varying slice thicknesses.*
> >
> > * **Potential for Adaptive Interpolation (Outlook):** While our experiments show that a standard linear progression of the Slerp weight works well, we acknowledge the potential for further refinement based on physical spacing. As defined in Eq. (10), the interpolation is controlled by the normalized index $\alpha_i \in [0, 1]$. In future work, this parameter $\alpha_i$ can be made adaptive to the physical slice thickness. For example, thicker slices (implying lower anatomical correlation) could utilize a non-linear mapping of $\alpha$ or a wider angular distance between anchors to reflect the reduced dependency. This inherent flexibility suggests that ISCS can be further optimized for extreme anisotropic cases, although our current "parameter-free" version already yields stable improvements. We will explore this direction in future work.
> >
> > ---
> >
> > **W-3 & Q-4. The paper lacks comparison with recent 3D-aware diffusion priors (e.g., DiffusionBlend [Song et al., NeurIPS '24]) or multi-plane consistency methods**
> >
> > **A:** We thank the reviewer for pointing out this important omission. We have now included comparisons with DiffusionBlend and TPDM. See `Q-1 in the Global Responses` at the beginning of the rebuttal, please.
> >
> > ---
> >
> > **W-4. Only Slerp vs. identical noise is analyzed. Additional ablations (e.g., interpolation degree, anchor distance, or adaptive correlation strength) would strengthen claims**
> >
> > **A:** See `Q-3 in the Global Responses` at the beginning of the rebuttal, please.
> >
> > ---
> >
> > **Q-2. How does ISCS interact with deterministic samplers (e.g., DDIM with η=0)?**
> >
> > **A:** See `Q-2 in the Global Responses` at the beginning of the rebuttal, please.
> >
> > ---
> >
> > **Q-5. Could Slerp interpolation be extended to spatially adaptive correlation where anatomical gradient drives?**
> >
> > **A:** We thank the reviewer for this inspiring suggestion. We agree that extending ISCS to a spatially adaptive regime is a promising and theoretically viable direction.
> >
> > * **Proposed Mechanism:** In principle, the scalar interpolation weight $\alpha_i$ in Eq. (10) could be expanded into a spatially varying coefficient map $\boldsymbol{\alpha}_i(x,y)$. This map could be modulated by anatomical priors (e.g., gradients estimated from a preliminary reconstruction). For instance, one could theoretically relax the correlation strength in regions with high-frequency anatomical transitions (e.g., lesion boundaries) to allow for greater stochastic diversity, while enforcing stronger correlation in homogeneous regions to suppress noise.
> > * **Future Scope:** While this "guided correlation field" offers exciting possibilities for fine-grained control, it introduces additional complexity (e.g., designing the mapping function from gradient to $\boldsymbol{\alpha}$) and moves away from our current parameter-free, plug-and-play design philosophy. Therefore, while we acknowledge the high potential of this idea, we consider its implementation beyond the scope of this paper and explicitly plan to explore it in our future work.
> >
> > ---

---

> ### Author Response · Authors · 2025-11-26
>
> **Q-7. Are there any cases where ISCS causes artifacts or over-correlation (e.g., repetitive patterns)?**
>
> **A:** In our experiments, we did not observe noticeable artifacts or repetitive patterns introduced by ISCS. We attribute this to the fundamental formulation of the inverse problem solver:
>
> * **Measurement-Driven Structure:** The anatomical structure is primarily governed by the measurement data via the data consistency step (Eq. 7). This step strictly enforces fidelity to the observed projections for each slice, preventing the "hallucination" of repetitive patterns that are not supported by the data.
> * **Role of ISCS:** ISCS constrains the stochasticity (the noise trajectory) rather than the solution itself. By narrowing the solution space to a smooth manifold, it eliminates high-frequency inter-slice jitter but does not impose a rigid structural template. This allows the data consistency term to freely dictate anatomical content, ensuring that genuine structural variations are preserved.
>
> ---
>
> **Q-8. Is the improvement consistent across random seeds, or does stochastic correlation introduce bias**
>
> **A:** The improvement provided by ISCS is consistent across random seeds and does not introduce significant bias. See `Q-3 in the Global Responses` at the beginning of the rebuttal, please.

---

### Official Review · Reviewer_4QC1 · 2025-11-01

**Soundness:** 3
**Presentation:** 3
**Contribution:** 3
**Rating:** 4
**Confidence:** 3

**Summary:**

The paper proposes Inter-Slice Consistent Stochasticity (ISCS) to improve 3D medical reconstructions obtained from 2D diffusion priors. The key idea is to synchronize the stochastic component of the reverse diffusion process across adjacent slices by initializing each slice’s noise via spherical linear interpolation (slerp) between two anchor noise maps. This smoothly correlates randomness across slices, aligning sampling trajectories without adding new loss terms, hyperparameters, retraining, or extra computation, and can be dropped into existing 2D-diffusion-based inverse-problem solvers (e.g., DDNM, DDS) as a plug-and-play component.

Experiments on three tasks: sparse-view CT, limited-angle CT, and MRI isotropic super-resolution—show consistent improvements in PSNR/SSIM and LPIPS over DDNM and DDS baselines, often rivaling or exceeding TV-regularized variants, while qualitatively reducing inter-slice artifacts in coronal/sagitta

**Strengths:**

- The method is simple, clear, and effective: replacing independent per-slice noise with a slerp-correlated noise volume improves inter-slice consistency without extra training or losses, and integrates directly into standard DDIM-style updates.

- Presentation is generally clear, with a helpful geometric interpretation (independent vs. identical vs. slerp noise) that motivates why smooth correlation avoids both uncorrelated flicker and over-rigid copying artifacts.

- Code is provided.

**Weaknesses:**

- Several related literature are missed. Di-Fusion (Wu etal., ICLR 2025) and DDM^2 (Xiang etal., ICLR 2023) and many more other papers on inverse problem solving for medical data. Those literature were published on ICLR and for a similar problem, even if not exactly the same. The authors are highly encouraged to discuss those related work in the paper.

- Lack of a dedicated inter-slice consistency metric in the main results. The quantitative tables report PSNR/SSIM/LPIPS only (per view). While the paper includes an “inter-slice difference” trajectory to illustrate coherence during sampling, this analysis is not integrated as a headline metric alongside PSNR/SSIM/LPIPS, making it hard to compare inter-slice consistency at a glance across methods and tasks. A specialized, standardized metric for inter-slice consistency would better highlight the core contribution.

- Insufficient theoretical and practical justification that slerp is the best interpolation choice. The current ablation contrasts identical noise (BCS) vs. slerp and shows slerp avoids “copying artifacts” and slightly improves quantitative metrics. However, other plausible interpolations (e.g., simple linear interpolation in noise space) are not evaluated, leaving open whether slerp is uniquely effective or merely one effective option.

- Comparative breadth. Although comparisons include strong diffusion-based solvers (DDNM, DDS), plus classical baselines (e.g., FDK, ADMM-TV), it would strengthen the evidence to compare against additional medical inverse-problem baselines that may (or may not) benefit from noise design, to more clearly position ISCS’s impact.

**Questions:**

- Training protocol parity: For the base diffusion models used with DDNM and DDS, were they derived from the same trained prior(s) (same data, architecture, and training protocol), with only the solver differing? A concise statement about identical priors across solvers would clarify fairness.

- Beyond inverse problems: Could ISCS plausibly extend to direct 3D volume generation (e.g., from very sparse or no views) by enforcing correlated stochasticity across slices during unconditional or weakly conditioned sampling? Any preliminary observations or limitations?

---

> ### Author Response · Authors · 2025-11-26
> **Official Comment by Authors**
>
> We thank the reviewer for the positive assessment of our work, highlighting that our method is "simple, clear, and effective." We also appreciate the constructive feedback regarding the related literature and evaluation metrics. We have carefully revised our paper to address these points. Below is our detailed response.
>
> ---
>
> **W-1: Several related literature are missed.**
>
> **A:** We thank the reviewer for pointing out these two DM-based works on medical imaging inverse problems. Our goal in this paper is to design a plug and play method for solving medical imaging inverse problems that use unconditional diffusion models as generative priors. In contrast, `Di-Fusion` and `DDM^2` focus on diffusion weighted MRI denoising and build task specific architectures and objectives based on diffusion model ideas. *We have added citations of these methods in the revised manuscript.*
>
> ---
>
> **W-2: Lack of a dedicated inter-slice consistency metric in the main results.**
>
> **A:** We agree that a specialized metric is essential for a holistic evaluation. In the revised manuscript, we have integrated a dedicated consistency metric into the main quantitative results (Table 1).
>
> We define SDiff as the mean absolute difference between adjacent slices:
> $$\text{SDiff} = \frac{1}{S-1} \sum_{i=1}^{S-1} \text{Mean}(|x_{i+1} - x_i|).$$
> To evaluate fidelity properly (avoiding the over-smoothing pitfall where $\text{SDiff} \to 0$), we report the Absolute Gap ($|\Delta|$) between the reconstruction and the Ground Truth:
> $$||\Delta|| = || \text{SDiff}\_{\text{recon}} - \text{SDiff}\_{\text{GT}} ||.$$
> Here, a smaller $|\Delta|$ indicates that the reconstructed volume successfully replicates the natural anatomical coherence of the GT.
>
> As shown in the updated Table 1 (and `Table R.1 in the supplementary material`), our method consistently achieves the lowest $|\Delta|$ compared to baselines. This confirms that ISCS not only reduces artificial flickering but also restores the 3D coherence to a level that closely matches the real anatomy.

---

> ### Author Response · Authors · 2025-11-26
> **Official Comment by Authors**
>
> **W-3. Insufficient theoretical and practical justification that slerp is the best interpolation choice.**
>
> **A:** We are extremely grateful to the reviewer for raising this question. It prompted us to conduct a deeper investigation into the interpolation mechanism, leading to unexpected and fascinating findings that have significantly enriched our understanding of diffusion-based inverse solvers.
>
> - **Theoretical Misalignment of Linear Interpolation:** First, we clarify the theoretical concern. As shown in `Fig. R. 1 in the supplementary material`, in high-dimensional spaces, samples concentrate on the hypersphere surface. Simple linear interpolation (LERP) moves through the interior of the sphere, reducing the vector magnitude. This does not merely violate a Gaussian assumption; more critically, it results in a severe mismatch between the injected noise magnitude and the diffusion model’s prescribed noise schedule at that specific timestep. Theoretically, this mismatch can disrupt the reverse process, as the model receives input with a "wrong" noise level, potentially causing artifacts or texture degradation [1].
> - **Experimental Observation (The "Surprise"):** Following your suggestion, we evaluated LERP and observed a dual phenomenon (as shown in `Fig. R. 2 in the supplementary material`):
>   - *Early Steps (High Noise)*: As expected theoretically, LERP performs poorly. The trajectory is unstable, and intermediate results are extremely noisy due to the significant noise level mismatch mentioned above. In contrast, Slerp maintains a stable trajectory with a progressive recovery from low to high frequencies.
>   - *Late Steps (Low Noise)*: Surprisingly, LERP begins to outperform Slerp in the final iterations. Our quantitative tracking shows that LERP yields cleaner final results. We hypothesize that in the final stages, where the noise schedule ($\sigma_t$) is already small, LERP further reduces the noise magnitude (effectively performing an "aggressive" denoising). The diffusion model appears to benefit from this reduced variance at the end, acting as a refiner to produce cleaner outputs.
> - **A "Hybrid" Strategy:** Based on this insight, we implemented a simple Hybrid Strategy: using Slerp for the majority of the process (to ensure trajectory stability and correct noise scheduling) and switching to LERP only in the final steps. This heuristic achieved better performance than using either Slerp or LERP alone.
>
> |          |       | LERP  | SLERP | SLERP+LERP |
> | -------- | ----- | ----- | ----- | ---------- |
> | **Axial**    | PSNR  | 37.19 | 36.56 | **37.33**  |
> |          | SSIM  | 0.926 | 0.921 | **0.928**  |
> |          | LPIPS | 0.048 | 0.048 | **0.048**  |
> | **Coronal**  | PSNR  | 37.44 | 36.84 | **37.58**  |
> |          | SSIM  | 0.928 | 0.927 | **0.932**  |
> |          | LPIPS | 0.059 | 0.059 | **0.057**  |
> | **Sagittal** | PSNR  | 37.68 | 37.08 | **37.82**  |
> |          | SSIM  | 0.924 | 0.922 | **0.928**  |
> |          | LPIPS | 0.057 | 0.059 | **0.056**  |
>
> While Slerp is the theoretically robust choice for maintaining consistency with the diffusion schedule (and thus our default), the behavior of LERP suggests that there is unexplored potential in manipulating the noise magnitude for inverse problems. While a full exploration of this "Hybrid Schedule" is beyond the scope of this paper, we have included these ablation results and the discussion in the revised Appendix. We sincerely thank the reviewer for this inspiring question, which has opened a valuable direction for our future work.
>
> > [1] Zheng, PengFei, et al. "Noisediffusion: Correcting noise for image interpolation with diffusion models beyond spherical linear interpolation." ICLR 2024.
>
> ---
>
> **W-4. Comparative breadth. Although comparisons include strong diffusion-based solvers (DDNM, DDS), plus classical baselines (e.g., FDK, ADMM-TV), it would strengthen the evidence to compare against additional medical inverse-problem baselines that may (or may not) benefit from noise design, to more clearly position ISCS’s impact.**
>
> **A:** We thank the reviewer for this comment. Our work focuses on improving 2D DM-based DIS from the perspective of noise design. ISCS is independent of the specific inverse solver and only interacts with the re-noising step. In the current manuscript, we primarily chose two representative methods, DDNM and DDS, due to computational and space limits. However, **as a supplement to further strengthen our evaluation, we have added comparisons with methods that learn 3D diffusion priors, specifically TPDM and DiffusionBlend, in the revised submissions**. The results across multiple tasks confirm that ISCS consistently improves 3D inter-slice consistency. We will clarify that ISCS can be broadly applied to other 2D DM-based medical inverse solvers. In future work, we plan to extend the experimental evaluation to additional solvers to further demonstrate this generality.

---

> > ### Author Response · Authors · 2025-11-26
> > **Official Comment by Authors**
> >
> > **Q-1. Training protocol parity: For the base diffusion models used with DDNM and DDS, were they derived from the same trained prior(s) (same data, architecture, and training protocol), with only the solver differing? A concise statement about identical priors across solvers would clarify fairness.**
> >
> > **A:** Yes, there is **no difference** in the underlying **diffusion prior** itself. For example, DDS and DDS+ISCS share the same pretrained diffusion prior and other settings; the **only** difference between them is the noise sampling strategy during inference (independent noise vs. ISCS). In Sec. 4.1 (Implementation Details) we describe the diffusion prior used in all methods. To avoid confusion, we will make it explicit that the training protocol for the diffusion model is strictly identical across all DM-based baselines and variants. We will add this clarification to Sec. 4.1 to make the fairness of the comparison clear.
> >
> > ---
> >
> > **Q-2. Beyond inverse problems: Could ISCS plausibly extend to direct 3D volume generation (e.g., from very sparse or no views) by enforcing correlated stochasticity across slices during unconditional or weakly conditioned sampling? Any preliminary observations or limitations?**
> >
> > **A:** We thank the reviewer for this insightful question. Our current work focuses on 3D inverse problems in which the measurements (for example, sinograms or low-resolution slices) provide strong constraints, and ISCS is used to shape the stochasticity of a 2D diffusion prior so that independently sampled slices form a coherent 3D volume. In other words, the proposed ISCS does not learn a 3D structural prior from data, but instead modifies the sampling noise of an already trained 2D model under a given measurement operator.
> >
> > - ***Outlook on Potential Directions.*** For this reason, directly applying ISCS to a pretrained 2D slice diffusion model for unconditional 3D volume generation is likely not sufficient. However, we see several promising extensions. One potential direction is to use ISCS as a tool during fine-tuning, similar in spirit to what has been done in video diffusion models [2]. During fine-tuning, one could inject ISCS-controlled noise for neighboring slices from the same volume, instead of independent noise, to encourage the model to internalize latent 3D priors across slices. We plan to explore this idea in future work.
> > - ***A Small Try.*** In addition, we have made a preliminary attempt to extend ISCS to a specific medical volume generation pipeline, GenerateCT [1]. GenerateCT first uses a 3D latent diffusion model in stage 1 to generate a coarse ($D \times 128^2$) volume, then applies a 2D diffusion based super-resolution model slice by slice in stage 2 to obtain a ($D \times 512^2$) volume. Because stage 2 operates independently per slice, the final volumes exhibit noticeable inter-slice discontinuities. When we simply apply ISCS to the sampling noise in stage 2, we observe a clear qualitative reduction of these discontinuities in the generated volumes, as shown in the qualitative visualizations in `Fig. R. 3 of the supplementary material`. While these observations are still preliminary and qualitative, they suggest that ISCS can be a useful ingredient for future work on 3D volume generation, especially when combined with appropriate training or fine-tuning schemes.
> >
> > > [1] Hamamci, Ibrahim Ethem, et al. "GenerateCT: Text-conditional generation of 3D chest CT volumes." ECCV 2024.
> > >
> > > [2] Liu, Chao, and Arash Vahdat. "EquiVDM: Equivariant Video Diffusion Models with Temporally Consistent Noise." arXiv preprint arXiv:2504.09789 (2025).

---

### Author Response · Authors · 2025-11-26
**Global Responses by Authors**

We sincerely thank the reviewers for their insightful comments and constructive suggestions!

We appreciate the reviewers' recognition of our method's simplicity and plug-and-play efficacy (4QC1, fFZa, 6tkB), strong geometric motivation (4QC1, PqvB), and extensive empirical superiority over TV regularization (PqvB, fFZa, 6tkB).

Here, we first address three common questions raised by multiple reviewers: (1) comparisons with 3D-aware diffusion priors, (2) applicability to deterministic samplers, and (3) the stability of the interpolation strategy (anchor selection). Subsequently, we provide detailed responses to the individual feedback provided by each reviewer.

---

**Q-1: Comparison with 3D-aware DM-based methods (PqvB, fFZa, 6tkB).**

**A:** We appreciate the suggestion to compare with explicit 3D-aware diffusion priors. We have conducted additional experiments with `DiffusionBlend` and `TPDM` on CT reconstruction tasks.

**(1) Experimental Setup for Fairness.** To ensure a fair comparison of the data prior, we re-trained both TPDM and DiffusionBlend priors using the exact same AAPM CT dataset (1mm slice thickness) used for our 2D baselines. We evaluated two distinct subjects to test generalization:

- Subject **L506**: Standard 1mm thickness ($300\times256\times256$). Matches the training distribution.
- Subject **L221**: Thick 5mm slices ($99\times256\times256$). Represents a domain shift challenge.

**(2) Adaptation of TPDM.** The original TPDM is constrained to cubic volumes (e.g., $256^3$). To adapt it for L506 ($D=300$), we extended TPDM with a sliding window approach along the coronal axis. However, TPDM strictly requires the slice depth $D \ge 256$ to function in its orthogonal view consistency check. Consequently, it cannot handle Subject L221 ($D=99$), resulting in missing values in our tables.

**(3) Results and Robustness Analysis.** The quantitative results are reported in `Table R1` (SVCT) and `Table R2` (LACT).

- **On Ideal Data (L506, 1mm):** 3D-aware methods (DiffusionBlend/TPDM) generally outperform 2D methods (DDS) and ISCS, confirming the power of explicit 3D priors when test data perfectly matches the training distribution (1mm). *However, ISCS significantly narrows the gap compared to the baseline DDS*.
- **On Challenging Data (L221, 5mm):** The advantage of 3D-aware methods diminishes. Since the priors were learned on 1mm thin slices, applying them to 5mm thick slices introduces a distribution mismatch. In some metrics (e.g., SSIM for SVCT), ISCS achieves comparable or even superior performance to DiffusionBlend. This demonstrates that ISCS, by regulating stochasticity rather than enforcing a rigid 3D spatial prior, *offers greater robustness and flexibility across varying acquisition protocols*.

|||**Axial**|||**Coronal**|||**Sagittal**|||
|-|-|-|-|-|-|-|-|-|-|-|
|**Subject**|**Method**|**PSNR**|**SSIM**|**LPIPS**|**PSNR**|**SSIM**|**LPIPS**|**PSNR**|**SSIM**|**LPIPS**|
|**L506**|DDS|34.76|0.919|0.069|35.12|0.906|0.149|35.33|0.904|0.141|
||DDS+ISCS|36.97|0.937|0.064|37.75|0.944|0.07|38.16|0.942|0.065|
||TPDM|37.59|0.944|0.063|38.4|0.95|0.068|38.64|0.948|0.062|
||DiffusionBlend|38.22|0.943|0.034|38.95|0.945|0.047|39.29|0.943|0.042|
|**L221**|DDS|37.83|0.952|0.045|37.96|0.952|0.055|38.62|0.95|0.045|
||DDS+ISCS|38.7|0.956|0.045|39.03|0.963|0.037|39.7|0.961|0.031|
||TPDM|-|-|-|-|-|-|-|-|-|
||DiffusionBlend|38.84|0.944|0.046|39.05|0.946|0.054|39.44|0.946|0.045|

*Table R1. Quantitative results of compared method in SVCT task of 30 views.*

|||**Axial**|||**Coronal**|||**Sagittal**|||
|-|-|-|-|-|-|-|-|-|-|-|
|**Subject**|**Method**|**PSNR**|**SSIM**|**LPIPS**|**PSNR**|**SSIM**|**LPIPS**|**PSNR**|**SSIM**|**LPIPS**|
|**L506**|DDS|29.03|0.885|0.086|29.84|0.828|0.197|29.13|0.829|0.194|
||DDS+ISCS|31.65|0.911|0.071|32.9|0.917|0.082|32.49|0.92|0.077|
||TPDM|30.95|0.912|0.062|32.48|0.92|0.07|32.49|0.924|0.064|
||DiffusionBlend|31.11|0.915|0.037|33.16|0.917|0.053|32.76|0.922|0.049|
|**L221**|DDS|29.15|0.895|0.053|29.94|0.86|0.139|30.12|0.862|0.124|
||DDS+ISCS|30.45|0.908|0.042|31.42|0.91|0.053|31.85|0.912|0.046|
||TPDM|-|-|-|-|-|-|-|-|-|
||DiffusionBlend|31.1|0.918|0.029|32.94|0.921|0.045|33.02|0.923|0.037|

*Table R2. Quantitative results of compared method in LACT task of [0, 100]°.*

**(4) Conclusion: Orthogonality and Efficiency.** While 3D-aware methods offer high performance, they come with task specific network designs, 3D-aware architectures, and additional training, which significantly increases the computational cost and engineering complexity.. ISCS focuses on a different axis: improving 2D priors through sampling-time noise correlation without architectural changes. Crucially, ISCS is orthogonal to 3D-aware methods. In principle, ISCS can be integrated into DiffusionBlend or TPDM to further enhance their consistency, which we plan to explore in future work.

---

> ### Author Response · Authors · 2025-11-26
>
> **Q-2: ISCS with deterministic sampler (e.g., DDIM with $\eta=0$) (PqvB, fFZa, 6tkB)**
>
> **A:** We appreciate the reviewers' interest in the interaction between ISCS and deterministic sampling. We address this from three perspectives: technical applicability, empirical improvement, and the role of stochasticity in inverse problems.
>
> **(1) Technical Applicability to Deterministic Samplers.** We clarify that ISCS is fully compatible with deterministic samplers (e.g., DDIM with $\eta=0$). Although the denoising trajectory is deterministic, it is entirely conditioned on the initial noise $x_T$. By applying ISCS to correlate the initial noise volume $x_T$ across slices, we ensure that the starting points of the reverse diffusion process lie on a spatially consistent manifold. This allows the subsequent deterministic steps to preserve inter-slice coherence. In principle, this strategy is applicable to any ODE-based sampler, including flow matching methods.
>
> **(2) Quantitative Improvement under Deterministic Settings.** To validate this, we conducted experiments on the SVCT-30 task using DDIM with $\eta=0$. As shown in `Table R3`, applying ISCS to the initial noise significantly boosts performance compared to the baseline using independent initialization. Notably, we observe substantial gains in SSIM and LPIPS for the Coronal and Sagittal views (e.g., Coronal LPIPS improves from 0.239 to 0.065), confirming that ISCS effectively mitigates slice-wise artifacts even in deterministic regimes.
>
> **(3) The Necessity of Stochasticity in Inverse Problems.** While ISCS is effective for $\eta=0$, our extensive ablations (reported in `Table R4`) indicate that deterministic sampling itself is suboptimal for this task. We observe a monotonic improvement in reconstruction quality as $\eta$ increases, with $\eta \approx 1$ (fully stochastic) yielding the best results. This aligns with the consensus in recent diffusion-based inverse problem literature [1-5], which suggests that stochasticity is essential for escaping local minima and recovering high-frequency textures. Therefore, while ISCS supports deterministic sampling, we adopt a stochastic setting as the default to maximize reconstruction fidelity.
>
> |              |       | **DDIM with $\eta=0$** | **DDIM with $\eta=0$ + ISCS** |
> | ------------ | ----- | ---------------------- | ----------------------------- |
> | **Axial**    | PSNR  | 32.30                  | **36.19**                     |
> |              | SSIM  | 0.864                  | **0.924**                     |
> |              | LPIPS | 0.065                  | **0.053**                     |
> | **Coronal**  | PSNR  | 32.67                  | **37.10**                     |
> |              | SSIM  | 0.806                  | **0.935**                     |
> |              | LPIPS | 0.239                  | **0.065**                     |
> | **Sagittal** | PSNR  | 32.93                  | **37.58**                     |
> |              | SSIM  | 0.805                  | **0.932**                     |
> |              | LPIPS | 0.228                  | **0.059**                     |
>
> *Table R3. Effect of applying ISCS to the initial noise $x_T$ under deterministic DDIM sampling $\eta = 0$) in SVCT task of 30 views.*
>
> | **$\eta$** | **PSNR**  | **SSIM**  |
> | ---------- | --------- | --------- |
> | **0.0**    | 34.48     | 0.916     |
> | **0.1**    | 34.56     | 0.918     |
> | **0.2**    | 34.77     | 0.923     |
> | **0.3**    | 35.06     | 0.927     |
> | **0.4**    | 35.38     | 0.932     |
> | **0.5**    | 35.70     | 0.936     |
> | **0.6**    | 35.99     | 0.939     |
> | **0.7**    | 36.26     | 0.942     |
> | **0.8**    | 36.52     | 0.945     |
> | **0.9**    | 36.77     | 0.948     |
> | **1.0**    | **37.08** | **0.951** |
>
> *Table R4. Ablation study on the selection of $\eta$ in DDIM sampler for SVCT task of 30 views.*
>
> > [1] Kwon, Taesung, and Jong Chul Ye. "Solving video inverse problems using image diffusion models." *ICLR 2025*.
> >
> > [2] Zhu, Yuanzhi, et al. "Denoising diffusion models for plug-and-play image restoration." *CVPRW 2023*.
> >
> > [3] Nie, Shen, et al. "The blessing of randomness: Sde beats ode in general diffusion-based image editing." *ICLR 2024*.
> >
> > [4] Wang, Yinhuai, Jiwen Yu, and Jian Zhang. "Zero-shot image restoration using denoising diffusion null-space model." *ICLR 2023*.
> >
> > [5] Kawar, Bahjat, et al. "Denoising diffusion restoration models." *NeurIPS 2022*.

---

> > ### Author Response · Authors · 2025-11-26
> >
> > **Q-3: Stability of Interpolation Strategy (Anchor Selection) (PqvB, fFZa, 6tkB).**
> >
> > **A:** We address concerns regarding the sensitivity of ISCS to the selection of anchor noise vectors ($z_1, z_S$) and random seeds. We provide both a theoretical justification and an empirical ablation study.
> >
> > **(1) Theoretical Stability: Concentration of Measure.** From a geometric perspective, the "randomness" of the angle between two high-dimensional noise vectors is strictly bounded. Due to the concentration of measure phenomenon, the angle between any two independent vectors sampled from a high-dimensional standard normal distribution concentrates tightly around 90 degrees (orthogonality) with negligible variance. This implies that our default random sampling strategy naturally yields a geometrically consistent interpolation path length across different runs, inherently minimizing instability.
> >
> > **(2) Empirical Ablation on Angular Distance.** To rigorously verify this robustness and rule out the existence of a **sensitive** "optimal spacing," we conducted a controlled ablation study. Instead of random sampling, we fixed $z_1$ and generated $z_S$ with a constrained angle $\theta \in \{30^{\circ}, 60^{\circ}, 90^{\circ}, 120^{\circ}, 150^{\circ}, 175^{\circ}\}$. We performed **10** independent runs for each angle setting on the SVCT-30 task.
> >
> > The quantitative results (reported in `Table R5`) reveal remarkable stability:
> >
> > - **Performance Consistency:** The reconstruction metrics fluctuate negligibly across the entire angular spectrum. For instance, the Axial PSNR varies by less than 0.15 dB between the minimum ($150^{\circ}$) and maximum ($175^{\circ}$) performance.
> > - **Low Variance across Seeds:** The standard deviations across the 10 runs are consistently low (mostly $< 0.2$ dB), indicating that the method is highly reproducible and insensitive to specific random initializations.
> >
> > These findings confirm that ISCS is robust to both anchor angle and random seed selection. Consequently, the standard practice of independent random sampling (yielding $\approx 90^{\circ}$) is validated as an optimal, parameter-free default choice that requires no further tuning.
> >
> >
> > | **Angle** | **Axial PSNR** | **Axial SSIM** | **Axial LPIPS** | **Coronal PSNR** | **Coronal SSIM** | **Coronal LPIPS** | **Sagittal PSNR** | **Sagittal SSIM** | **Sagittal LPIPS** |
> > | --------- | -------------- | -------------- | --------------- | ---------------- | ---------------- | ----------------- | ----------------- | ----------------- | ------------------ |
> > | **30°**   | 36.84 ± 0.14   | 0.937 ± 0.001  | 0.064 ± 0.003   | 37.66 ± 0.14     | 0.944 ± 0.001    | 0.068 ± 0.003     | 38.04 ± 0.15      | 0.942 ± 0.001     | 0.063 ± 0.003      |
> > | **60°**   | 36.83 ± 0.13   | 0.937 ± 0.001  | 0.064 ± 0.002   | 37.65 ± 0.15     | 0.944 ± 0.001    | 0.067 ± 0.002     | 38.06 ± 0.16      | 0.942 ± 0.001     | 0.063 ± 0.002      |
> > | **90°**   | 36.84 ± 0.07   | 0.937 ± 0.000  | 0.066 ± 0.002   | 37.69 ± 0.08     | 0.945 ± 0.000    | 0.069 ± 0.002     | 38.07 ± 0.09      | 0.943 ± 0.000     | 0.065 ± 0.002      |
> > | **120°**  | 36.86 ± 0.05   | 0.937 ± 0.000  | 0.067 ± 0.001   | 37.69 ± 0.07     | 0.945 ± 0.000    | 0.069 ± 0.001     | 38.09 ± 0.08      | 0.943 ± 0.000     | 0.065 ± 0.001      |
> > | **150°**  | 36.79 ± 0.20   | 0.936 ± 0.002  | 0.065 ± 0.004   | 37.61 ± 0.23     | 0.944 ± 0.002    | 0.069 ± 0.004     | 38.00 ± 0.24      | 0.942 ± 0.002     | 0.064 ± 0.003      |
> > | **175°**  | 36.90 ± 0.04   | 0.938 ± 0.000  | 0.065 ± 0.001   | 37.74 ± 0.03     | 0.945 ± 0.001    | 0.069 ± 0.002     | 38.12 ± 0.06      | 0.943 ± 0.001     | 0.064 ± 0.001      |
> >
> > *Table R5. Quantitative results (Mean $\pm$ Std) of ISCS under different anchor angles (10 independent runs each) on SVCT-30 task.*

---

### Author Response · Authors · 2025-12-04
**Author Summary of Rebuttal Progress and Manuscript Updates**

We thank the AC for overseeing this submission. Given the recent adjustments in the review process, we provide this summary to outline the substantial progress made during the rebuttal period and the resulting consensus among reviewers.

---

**Summary of Score Changes & Reviewer Consensus:**
During the discussion phase, we engaged in an extensive and constructive dialogue with all reviewers. This process led to a significant convergence in positive assessments:
* **Reviewer `fFZa` (Major Turnaround):** Upgraded score from **2 (Reject) $\to$ 8 (Accept)**. Following our clarifications and new experiments, the reviewer explicitly stated: *"The authors' rebuttal and additional experiments have clearly addressed all of my concerns... As there remain no substantive concerns that would justify rejecting the paper, I am significantly increasing my score."* **It is worth noting that both the score revision and the discussion took place before the leakage incident occurred.**
* **Reviewer `6tkB`:** Engaged positively, acknowledging the practical value of our method as a *"good default choice for practitioners"* and recognizing the effectiveness of our added baselines and theoretical clarifications.
* **General Consensus:** We believe we have also fully addressed the concerns of **Reviewer `PqvB`** (who initially gave a positive score of 6) and **Reviewer `4QC1`** through our comprehensive new benchmarks.

---

**Key Actions Taken to Resolve Concerns:**

**1. Comparison with State-of-the-Art 3D-Aware Priors (Addressing `PqvB`, `fFZa`, `6tkB`)**
* **Action:** We conducted rigorous comparisons with methods that strictly learn 3D priors (**TPDM** and **DiffusionBlend**).
* **Outcome:** We demonstrated that while explicit 3D priors perform well in ideal settings, our method (ISCS) exhibits **superior robustness** on out-of-distribution data (e.g., varying slice thicknesses) and achieves competitive high-fidelity results without the high cost of 3D training.

**2. Robustness on Pathological & Heterogeneous Data (Addressing `PqvB`)**
* **Action:** To address concerns about potential over-smoothing of medical anomalies, we added experiments on the **DeepLesion** dataset and data with **varying slice thicknesses** (3mm, 5mm, 7.5mm).
* **Outcome:** Results (Fig. R5 in Supp) confirm that ISCS preserves critical pathological details (e.g., small lesions) significantly better than TV regularization while maintaining consistency across diverse acquisition protocols.

**3. Theoretical Clarifications & Deterministic Sampling (Addressing `fFZa`, `6tkB`, `4QC1`)**
* **Action:** We clarified the theoretical compatibility of ISCS with deterministic samplers (e.g., DDIM, $\eta=0$) and provided a **variance reduction analysis** to explain convergence.
* **Outcome:** We justified the geometric necessity of Slerp over linear interpolation using the high-dimensional concentration of measure phenomenon, solidifying the theoretical grounding of our approach.

**4. Comprehensive Metric & Baseline Expansion (Addressing `4QC1`, `6tkB`)**
* **Action:** We introduced a dedicated inter-slice consistency metric (**Absolute Gap of Slice Difference, $|\Delta|$**) and added comparisons with Video Diffusion methods (**PYoCo**) to highlight domain distinctions.

**Resolution of Individual Concerns:**
Beyond these common themes, we have meticulously addressed every independent question raised by individual reviewers, ranging from specific hyperparameter ablations (e.g., anchor stability) to clarifications on related works in video generation.

---

We sincerely thank the reviewers and the AC for their time and effort. The rigorous rebuttal process has not only resolved outstanding concerns but has also significantly elevated the quality and clarity of our manuscript. We believe the updated work now presents a robust, theoretically sound, and practically valuable contribution that aligns well with the high standards of the ICLR community.

---

### Meta-Review · Area_Chair_ZNSu · 2026-01-06

**Summary:**

My read is that this submission sits right on the borderline in terms of novelty, but it clears the bar on practical impact and0 "community usefulness," especially because it offers a genuinely low-friction improvement that many practitioners can adopt immediately.

Across reviewers, the central positive was consistent: ISCS is a simple, plug-and-play modification to 2D diffusion-prior inverse solvers that improves 3D inter-slice coherence without retraining, new losses, or noticeable overhead, and it empirically competes with (and often improves upon) the common TV-style inter-slice regularization. The main doubts were also consistent: (i) novelty could be incremental relative to temporally correlated noise work in video diffusion; (ii) the original evaluation under-emphasized the core claim (inter-slice consistency) due to lacking a dedicated metric in the main tables; (iii) questions on whether SLERP is uniquely necessary vs simpler interpolations (LERP), and how sensitive the method is to anchor choice / random seeds; (iv) breadth/positioning concerns, including missing comparisons to 3D-aware priors and to representative correlated-noise baselines; (v) questions about deterministic samplers and the broader relevance if one uses ODE-like or deterministic sampling; and (vi) generalization worries for pathological or heterogeneous acquisition settings (thick slices, anisotropic spacing, abrupt anatomical changes).

**Reviewer Concerns:**

The rebuttal substantially strengthened the paper on exactly these axes: it added stronger comparative baselines (3D-aware priors and a representative video-noise method), introduced a dedicated inter-slice consistency metric into the main results, clarified compatibility with deterministic sampling by correlating the initial noise, provided stability/seed and anchor-angle analyses (with low variance), expanded robustness evaluation to heterogeneous slice thickness and pathology (e.g., lesion preservation vs TV oversmoothing), and improved the theoretical positioning (variance reduction intuition + geometric argument for SLERP; plus an informative SLERP/LERP + hybrid observation rather than claiming SLERP is the only reasonable choice). As a result, the remaining issues are now mostly about "degree of novelty" and "how much theory is enough," rather than about correctness or missing evidence.

Given the improved experimental support, clearer positioning (as stochasticity control for 2D-prior-based 3D inverse problems), and the strong likelihood of real adoption, I lean slightly toward accept despite the borderline novelty.

**Reviewer Scores:**

The rebuttal plausibly converts this from "mixed borderline" to "accept-leaning borderline," mainly on practical value and strengthened evidence, with remaining debate centered on novelty/theory expectations rather than missing validation.

---

### Decision · Program_Chairs · 2026-01-26

Accept (Poster)